# UniBP: Toward Universal Backdoor Purification via Fine-Tuning

## Abstract

Deep neural networks (DNNs) remain vulnerable to backdoor attacks, perpetuating an arms race between attacks and defenses. Despite their efficacy against classical threats, mainstream defenses often fail under more advanced, defense-aware attacks, particularly clean-label variants that can evade decision-boundary shifting and neuron-pruning defenses. We present `UniBP`, a universal post-training defense that operates with only 1% of the original training data and unveils the relationship between batch normalization (BN) behavior and backdoor effects. At a high level, `UniBP` scrutinizes BN layers' affine parameters and statistics using a small clean subset (i.e., as small as 1% of the training data) to find the most impactful affine parameters for reactivating the backdoor, then prunes them and applies masked fine-tuning to remove the backdoor effects. We compare our method against 9 SOTA defenses, 9 backdoor attacks, and various attack/defense conditions, and show that `UniBP` consistently reduces the attack success rate from more than 90% to less than 5% while preserving clean performance, whereas other baselines degrade under smaller fine-tuning sets or stronger poisoning techniques. Our code is publicly available at `https://anonymous.4open.science/r/UniBP-BackdoorPostDefense/README.md`.

## 1 Introduction

Deep neural networks (DNNs) have achieved remarkable success across a wide range of applications, including image classification, speech recognition, and natural language processing (Mienye & Swart, 2024; Samek et al., 2021; Noor & Ige, 2025). However, their vulnerability to backdoor attacks has raised serious concerns about their robustness in security-critical settings (Li et al., 2022; 2023c; Zhang et al., 2024; Wan et al., 2024; Cheng et al., 2025). In a backdoor attack, an adversary injects malicious patterns, which are commonly referred to as triggers into the training data. As a result, the model performs normally on clean inputs but misclassifies inputs containing the trigger in a controlled manner.

**Backdoor attacks.** Backdoor strategies have continued to evolve, becoming increasingly stealthy and effective. Early dirty-label methods such as BadNets (Gu et al., 2019) poison both inputs and labels, while later attacks like WaNet (Nguyen & Tran, 2021) apply subtle, visually faithful transformations that embed nearly-invisible triggers. More recent adaptive variants, including COMBAT (Huynh et al., 2024) and SBL (Sequential Learning Generates Resilient Backdoors) (Pham et al., 2024a), are explicitly crafted to bypass existing defenses, for example, by operating in clean-label regimes or by manipulating training dynamics to produce resilient, detection-aware backdoors. These advancements challenge traditional defense paradigms.

**Defenses.** In response, the literature spans adversarial training, input sanitization, and post-training defense. Recent methods increasingly focus on post-training approaches due to their practicality in the era of transfer learning, where the training phase remains unmodified (Min et al., 2024; Lin et al., 2024). Representative methods include Neural Cleanse (Wang et al., 2019) and STRIP (Gao et al., 2019), which serve as detection-based post-training defenses: Neural Cleanse reverse-engineers class-wise minimal triggers to expose anomalies, while STRIP perturbs inputs and measures prediction entropy to detect triggered samples at inference. More recent purification defenses such as NAD (Li et al., 2021c), I-BAU (Zeng et al., 2021), ANP (Wu & Wang, 2021), FST (Min et al., 2024), and Unit Cheng et al. (2024) aim to handle a broader range of attacks using clean datasets. These methods

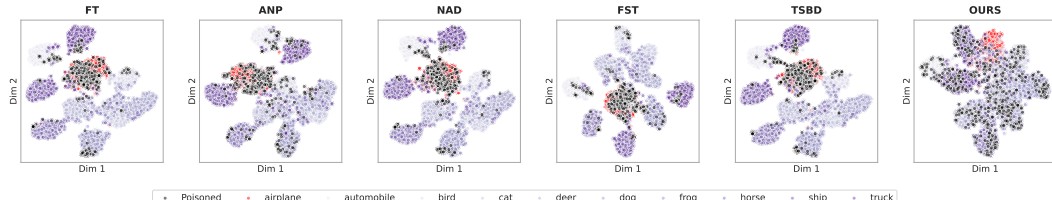

Figure 1: **t-SNE of feature embeddings on CIFAR-10.** Projection of penultimate-layer features for the backdoored model (PRETRAINED) and after applying defenses (ANP, FST, FT, NAD, TSBD, and UniBP). Clean samples are colored by class; poisoned samples are shown in black. All baselines fail in disrupting the overlapping representation of the backdoored data and the clean data of the targeted class (red).

respectively distill clean behavior from a teacher network (NAD), unlearn backdoors via a minimax objective on a small clean set (I-BAU), prune adversarially sensitive neurons (ANP), fine-tune to shift the representation of the backdoored model (FST), and tighten activation magnitudes to suppress anomalous trigger-induced activations (Unit).

However, these defenses primarily target dirty-label attacks and often rely on strong assumptions about the adversary's behavior, such as the use of a universal trigger Zeng et al. (2021) or significant activation shifts between backdoored and clean data Cheng et al. (2024); Zheng et al. (2022b). These assumptions are fundamentally violated by advanced clean-label attacks. For instance, Narcissus and Refool craft imperceptible triggers that produce minimal activation differences from clean samples, while COMBAT explicitly optimizes triggers to be inseparable from legitimate features of the target class in the feature space(cf. Figure 1). This vulnerability is further exacerbated under realistic constraints such as limited clean reference data and diverse attack configurations, where the defenders' ability to establish reliable decision boundaries becomes severely compromised.

**Our approach.** In this paper, we present a universal and practical *post-training* defense grounded in a key observation: Batch Normalization (BN) layers encode distributional statistics of both clean and poisoned data, and backdoor behavior exploits these statistics to steer specific activation pathways. While existing defenses such as BNP Zheng et al. (2022b) leverage BN statistics by comparing stored statistics of the backdoored model against running statistics computed on clean data, this direct comparison approach is inherently unstable—it operates on the *same* backdoored model where both sets of statistics are already contaminated, making it ineffective against sophisticated attacks such as clean-label backdoors where poisoned and clean distributions are carefully aligned.

Our method takes a fundamentally different approach by *simulating* the backdoor learning process itself. Specifically, we (i) *rectify and align* the BN statistics of a reinitialized clean model toward those of the backdoored model during fine-tuning, using the Fisher Information Matrix (FIM) to identify which affine parameters are most responsible for reproducing backdoor-specific activation patterns, then (ii) *selectively reset* only the targeted subset of critical BN affine parameters rather than all normalization layers, and (iii) apply *masked gradient fine-tuning* to prevent reactivation by malicious triggers while preserving model utility on clean inputs. This yields effective purification of pretrained models *without* prior knowledge of attack type, trigger pattern, or poisoned sample locations, operating with minimal clean data and assumptions. In practice, our procedure demonstrates effectiveness across various backdoor attacks and stability across diverse attack scenarios.

To summarize, our main contributions are as follows: (1) We unveil the relationship of BN layers' affine parameters and statistics toward the backdoor effect, and show that only a subset (i.e., 0.01%) of these parameters can sustainably disrupt the backdoor's attack success rate. We then introduce UniBP, a post-training defense that finds these affine parameters, then conducts pruning and masked fine-tuning to remove the backdoor from a poisoned model. (2) We empirically show that 9 fine-tuning defenses are often ineffective and unstable across major backdoor families. In contrast, our method is universal in that it is consistently effective against nine diverse backdoor attacks: traditional (BadNets (Gu et al., 2019), WaNet (Nguyen & Tran, 2021), Input-aware (Nguyen & Tran, 2020)), clean-label (LC (Turner et al., 2019), Narcissus (Zeng et al., 2023), Refool (Qi et al., 2023)), and adaptive (COMBAT (Huynh et al., 2024), SBL (Pham et al., 2024b), Adaptive Patch (Qi et al., 2023)) attack families. (3) We rigorously evaluate UniBP across a swath of attack settings and model architectures. We show that UniBP (i) preserves clean accuracy while maintaining stability and

resilience against each attack, (ii) requires only a small amount of clean data, and (iii) requires *no* assumptions about the implanted backdoor.

## 2 RELATED WORKS

### 2.1 BACKDOOR ATTACKS

Backdoor attacks aim to mislead a victim model into predicting a target label when a trigger is present in the input while maintaining unchanged performance on clean data. Backdoor attacks are categorized into *dirty-label* (Chen et al., 2017; Li et al., 2021b; Wang et al., 2022) and *clean-label* (Barni et al., 2019; Ning et al., 2021; Zeng et al., 2023) based on whether the trigger changes the underlying ground-truth label of the poisoned image. In *dirty-label backdoor attacks*, attackers modify both the image and its label. While the seminal BadNets (Gu et al., 2019) uses a single patch or pattern of bright pixels as a trigger, later works focus on making triggers less detectable through techniques such as image warping (Nguyen & Tran, 2021), input-aware dynamic triggers Nguyen & Tran (2020), or blended perturbations (Chen et al., 2017). However, the label-image inconsistency in dirty-label attacks remains visually detectable by humans upon careful inspection. In *clean-label backdoor attacks* (Barni et al., 2019; Ning et al., 2021; Zeng et al., 2023), triggers are only added to samples already belonging to the target class, eliminating label inconsistency and making detection significantly harder. LC (Turner et al., 2019) crafts adversarial perturbations to ensure the model associates the trigger with the target class. COMBAT (Huynh et al., 2024) learns an effective trigger generator through alternating optimization of the generator and a surrogate model, producing triggers that are difficult to separate from legitimate target-class features. Narcissus Zeng et al. (2023) generates instance-specific noise patterns optimized to be imperceptible while maintaining high attack success rates. Refool Liu et al. (2020) leverages class-wise feature representations to craft natural-looking triggers. Beyond these traditional attack paradigms, recent *adaptive attacks* are explicitly designed to evade specific defenses. SBL (Pham et al., 2024b) improves backdoor resilience against fine-tuning by trapping the model in sharp minima within the backdoored loss landscape via continual learning techniques. Adaptive Patch Qi et al. (2023) specifically targets embedding-based detection methods by optimizing triggers to minimize distributional shifts in feature space, making separation-based defenses ineffective.

### 2.2 BACKDOOR DEFENSES

In response to the growing threat of backdoor attacks, various defensive techniques have been proposed that operate during two stages of model training: (1) *training-stage* and (2) *post-training* defenses. *Training-stage defenses.* (Huang et al., 2022) aim to train a clean model even when the training data has been poisoned by an attacker. ABL (Li et al., 2021b) first isolates the backdoored data and then unlearns the isolated data using gradient ascent. D-ST/D-BR (Chen et al., 2022) leverages the insight that poisoned data are more sensitive to transformation compared to clean data, so they train a secure model from scratch or unlearn poisoned samples in a backdoored model. *Post-training defenses.* (Zheng et al., 2022a; Chen et al., 2018; Nguyen et al., 2024) aim to mitigate the backdoor effect on a poisoned model using a small set of known-clean data, typically achieved through pruning or fine-tuning. ANP (Wu & Wang, 2021) prunes sensitive neurons under adversarial neuron perturbation, as they are likely to be related to the injected backdoor. NAD (Li et al., 2021c) introduces an attention distillation method which uses a teacher network to guide the fine-tuning of the backdoored network. I-BAU Zeng et al. (2021) formulates backdoor unlearning as a minimax optimization problem, using a small clean validation set to isolate and unlearn backdoor-specific features while preserving model utility. RNP Li et al. (2023a) employs reconstructive neuron pruning based on the assumption that backdoor-related neurons exhibit distinct activation patterns, using sparsity constraints during clean data unlearning to identify and prune these neurons. FST (Min et al., 2024) encourages discrepancy between the fine-tuned model and the original model to achieve feature shifts that disrupt backdoor pathways. TSBD (Lin et al., 2024) leverages the insight that neuron weight changes are highly correlated between poisoned unlearning and clean unlearning, and proposes to (1) reinitialize neurons based on weight changes, and (2) fine-tune the model based on neuron activeness. PBP (Nguyen et al., 2024) first generates a neuron mask, then uses masked gradient optimization to eliminate backdoor effects.

BNP Zheng et al. (2022b) and BNA Li et al. (2025) are the closest methods to ours, leveraging batch-normalization statistics to detect and mitigate backdoors. BNP computes the KL divergence between the stored running statistics of a (potentially) backdoored model and those recomputed on clean data within the same model to identify suspicious layers, then resets the normalization parameters. However, this direct comparison on a single contaminated model can fail under sophisticated attacks, such as clean-label backdoors, where poisoned and clean activations are deliberately aligned. BNA, in turn, constructs a poisoned dataset using a reversed trigger and explicitly exploits the distributional shift between clean and triggered activations at each neuron by minimizing their KL divergence, but it relies on an estimated trigger, which may not be available or reliable in practice. More recently, Unit Cheng et al. (2024) proposes to tighten activation magnitudes based on the assumption that backdoor triggers induce anomalously large activations in specific channels. By constraining these activation ranges during fine-tuning, Unit aims to suppress backdoor pathways while maintaining clean accuracy. Despite these advances, current state-of-the-art defenses have not effectively tackled recently proposed resilient backdoor attacks, including SBL and COMBAT, underscoring the need for more robust defense mechanisms. Our method addresses this gap by taking a fundamentally different approach: rather than directly comparing statistics on the same backdoored model, we simulate the backdoor learning process itself to identify which parameters are most responsible for encoding backdoor behavior, enabling more precise and effective mitigation.

## 3 METHODOLOGY

### 3.1 PROBLEM STATEMENTS

Backdoor attacks often occur during model training (Gu et al., 2017; Zheng et al., 2022b;a; Wang et al., 2023), but modern ML workflows such as MLaaS platforms, transfer learning, and model marketplaces give users no control over this phase. Users acquire pre-trained models from third parties without visibility into training data or procedures, creating a fundamental asymmetry: attackers poison during training while defenders can only intervene post-hoc with limited clean data. Since backdoored models maintain clean accuracy indistinguishable from legitimate models, standard validation cannot detect compromise. We adopt a *post-training defense* setting where defenders receive a potentially backdoored model and possess only a small clean dataset $\mathcal{D}_{ft}$ for fine-tuning (Li et al., 2023b; 2021c). This reflects practical constraints where original training data is unavailable due to proprietary restrictions or privacy regulations. The objective is to eliminate backdoor behavior while preserving clean performance under severe data limitations.

**Attacker's goals.** Similar to most backdoor poisoning settings, we assume the attacker's goal is to alter the training procedure by using a small poisoned set, such that the resulting trained backdoored classifier, $f_{\theta^*}$, differs from a cleanly trained classifier. An ideal $f_{\theta^*}$ has the same response to clean samples, whereas it generates an adversarially chosen prediction, $\tau(y)$, when applied to backdoored inputs, $\varphi(x)$. **Defender's goal.** In contrast to the attacker, the defender—who has full access to the poisoned model $f_{\theta^*}$ and a limited benign fine-tuning set $\mathcal{D}_{ft}$ to get a clean/purified model $f_{\hat{\theta}}$ must (1) remove backdoors from $f_{\theta^*}$ to ensure correct behavior on triggered inputs and (2) preserve the model's performance on normal inputs during purification. In this work, following related post-training defenses Min et al. (2024); Wang et al. (2023); Lin et al. (2024), we adopt the following assumptions in a compact form: (i) the defender has no information about the backdoor trigger or the adversary's accessibility (e.g., poisoning rate, insertion mechanism), and we make no assumptions about any trigger/watermark; (ii) the defender has no access to the original training procedure and cannot obtain the full training dataset to retrain a new model; and (iii) the defender can collect or access a small, clean dataset representative of the training distribution (covering all classes), and may combine it with any available portion of the training data. This setting aligns with common post-training defenses (Min et al., 2024; Wang et al., 2023).

### 3.2 RELATIONSHIP OF BN LAYERS AND BACKDOOR EFFECT.

> **Finding 3.1: Backdoors shift BatchNorm (BN) statistics and affine parameter distributions**
>
> Training with a backdoor induces consistent, layer-dependent shifts in BN running means/variances and alters the distribution of BN affine parameters ($\gamma$, $\beta$) relative to clean baselines.

BatchNorm layers are often used in deep neural networks for the purposes of stabilizing and accelerating training (by reducing internal covariate shift), permitting larger learning rates, improving generalization via a mild regularization effect, and offering per-channel control through learnable affine parameters. Given a mini-batch of feature maps $x_{n,c,h,w}$ with batch size $N$ and spatial size $H \times W$, BN computes:

$$\mu_c = \frac{1}{NHW} \sum_{n,h,w} x_{n,c,h,w}, \quad \sigma_c^2 = \frac{1}{NHW} \sum_{n,h,w} \left(x_{n,c,h,w} - \mu_c\right)^2, \quad \hat{x}_{n,c,h,w} = \frac{x_{n,c,h,w} - \mu_c}{\sqrt{\sigma_c^2 + \varepsilon}}$$
(1)

and outputs the affine-transformed activations as $y_{n,c,h,w} = \gamma_c \hat{x}_{n,c,h,w} + \beta_c$, where $\gamma_c$ and $\beta_c$ are learned *affine* (scale/shift) parameters for channel $c$, and $\varepsilon$ ensures numerical stability. During training, $(\mu_c, \sigma_c^2)$ are computed from the current mini-batch while exponential moving averages are accumulated; at inference, these running estimates replace batch statistics. Our key insight (see 3.1) is that BN layers encode the training distribution via their running moments and affine parameters Zheng et al. (2022b); Li et al. (2024); Nguyen et al. (2024), and inserting a backdoor unavoidably shifts the distribution of the BN layers' statistic and affine parameters (see Figure 2a). Building on this observation, we articulate our second finding (3.2), which is central to our methodology: backdoor activation is governed by a small subset of BN affine channels; consequently, identifying and selectively editing these channels serves as an surprisingly effective lever for backdoor mitigation (cf. Figure 2c).

> **Finding 3.2: Backdoor activation is bottlenecked by a sparse subset of BN affine parameters**
>
> **Claim.** A small fraction of BN affine channels ($\gamma, \beta$) disproportionately governs trigger activation; selectively perturbing or resetting these top-ranked channels sharply reduces ASR with minimal impact on clean accuracy.

### 3.3 UNIBP: DETAILED DESCRIPTION

**High-Level Idea.** Motivated by the two findings mentioned above, we introduce a defense method including four components. (i) *batch-norm affine reset* to create an initialized model $\theta'$ from the backdoored model $\theta^*$; (ii) *affine mask calculation* by calculating FIM while the initialized model is trained with rectification to align the BN stats with the backdoored model; (iii) this mask will be used to prune the corresponding highly influential neurons to remove the backdoor effect, achieving a pruned model $\theta^u$; (iv) this pruned model is then fine-tuned using masked-gradient training with a clean dataset to achieve the purified version $\hat{\theta}$.

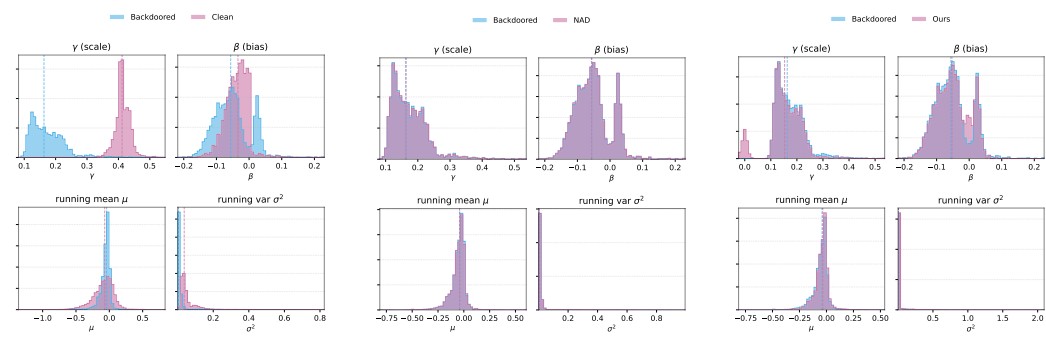

(a) Clean vs. Backdoored Models     (b) Backdoored vs. NAD Models     (c) Backdoored vs. Our Models

Figure 2: BatchNorm statistics $(\mu, \sigma)$ and affine parameters $(\gamma, \beta)$ for four model variants—*clean*, *backdoored*, *NAD* (Li et al., 2021a), and *Ours*—illustrating how backdoor training and purification affect BN layers. NAD leaves the backdoored BN statistics largely unchanged, whereas our method slightly shifts them while successfully removing the backdoor. ASR: clean $0.67\%$, backdoored $80.66\%$, NAD $78.66\%$, Ours $7.04\%$ (lower is better).

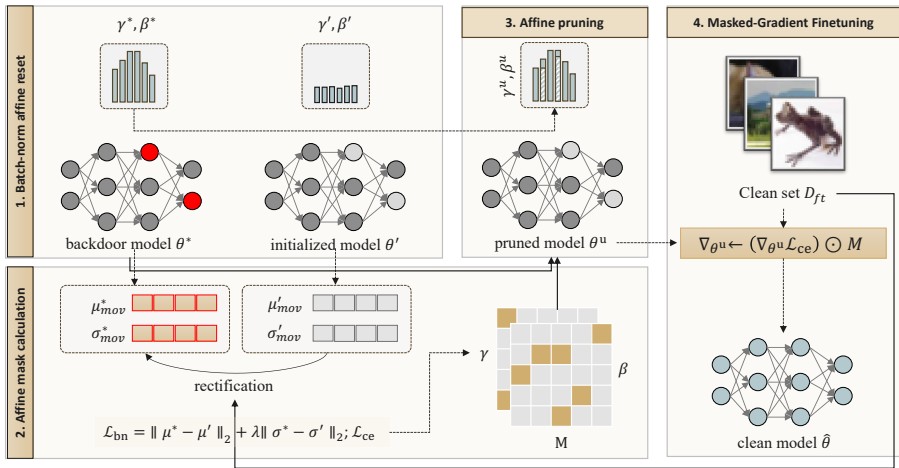

Figure 3: **UniBP** includes four phases. (1) *Batch-norm affine reset* reinitializes $\gamma, \beta$ of the backdoored model $\theta^*$ to obtain $\theta'$. (2) *Affine-mask calculation* rectifies BN moving statistics ($\mu_{\text{mov}}, \sigma_{\text{mov}}$) and learns a selection mask $M$ via the BN rectification loss $\mathcal{L}_{\text{bn}} = \|\mu' - \mu\|_2 + \lambda\|\sigma' - \sigma\|_2$ (with $\mathcal{L}_{\text{ce}}$). (3) *Affine pruning* removes suspect channels/affines, yielding $\theta^u$ with $\gamma^u, \beta^u$. (4) *Masked-gradient finetuning* on a small clean set $D_{ft}$ updates only unmasked parameters $(\nabla_{\theta^u}\mathcal{L}_{\text{ce}}) \odot M$, producing the purified model $\hat{\theta}$.

**Batch-norm affine reset.** Given a backdoored model $\theta^*$, we obtain the corresponding *re-initialized* model $\theta'$ by resetting BatchNorm affine parameters. Let $\mathcal{B}$ be the set of BN layers in $\theta^*$, and for each $\ell \in \mathcal{B}$ with $C_\ell$ channels let $(\gamma_\ell, \beta_\ell) \in \mathbb{R}^{C_\ell} \times \mathbb{R}^{C_\ell}$ denote its affine parameters (if present). For fixed reinit constants $(\gamma_0, \beta_0)$ (i.e., which are set default as $(1, 0)$), we define the operator $\mathcal{R}_{\text{BN}}$:

$$\theta' = \mathcal{R}_{\text{BN}}(\theta^*; \gamma_0, \beta_0), \quad (\gamma'_\ell, \beta'_\ell) = \begin{cases} (\gamma_0 \mathbf{1}_{C_\ell}, \beta_0 \mathbf{1}_{C_\ell}) & \text{if the BN layer } \ell \text{ has affine parameters,} \\ (\gamma_\ell, \beta_\ell) & \text{otherwise.} \end{cases}$$

**Affine Mask Calculation.** From the initialized model $\theta'$, we compute an *importance score* for each BN affine parameter that quantifies its contribution to rectifying the BatchNorm statistics of $\theta'$ ($\mu'_\ell, v'_\ell$) toward those of the backdoored model ($\mu^*_\ell, v^*_\ell$). This procedure mimics the alignment in which the statistics induced by a small clean fine-tuning set $\mathcal{D}_{\text{ft}}$ are drawn toward the mixed (clean and poisoned) distribution used to train $\theta^*$. To achieve this goal, we fine-tune the reinitialized model $\theta'$ by minimizing the rectification objective, and we quantify per-parameter importance via the (empirical) Fisher information computed on $\mathcal{D}_{\text{ft}}$. Specifically, we use $\mathcal{L}_{\text{rectify}}$ for optimization and estimate the diagonal Fisher for each parameter $\phi_i$ as in equation 3.

Let $\mathcal{B}$ be the set of BN layers, for each $\ell \in \mathcal{S}$, let $(\mu'_\ell, u'_\ell)$ denote the per-channel batch mean/variance computed on the current mini-batch as in Equation 1, and let $(\mu^*_\ell, v^*_\ell)$ be the corresponding references from the backdoored model. We define the per-layer deviation loss function as follows:

$$\mathcal{L}_{\text{BN}}^{(\ell)} = \left\|\hat{\mu}_\ell - \mu^*_\ell\right\|_2 + \lambda\left\|\hat{v}_\ell - v^*_\ell\right\|_2, \qquad \lambda = 0.05.$$

Then, the BN regularizer is calculated as: $\mathcal{L}_{\text{BN}} = \frac{1}{|\mathcal{S}|}\sum_{\ell \in \mathcal{S}} \mathcal{L}_{\text{BN}}^{(\ell)}$. This regularizer encourages the network's intermediate distributions to align with the reference (backdoored) normalization statistics, stabilizing activations without directly constraining $(\gamma, \beta)$. We then define the rectification objective by:

$$\mathcal{L}_{rectify} := \mathcal{L}_{CE}(x, y) + \log \mathcal{L}_{\text{BN}}. \tag{2}$$

Let $\Theta$ denote all trainable parameters and $\Theta_{\text{BN}} \subset \Theta$ the set of BN affine entries $\{\gamma_{\ell,c}, \beta_{\ell,c} : \ell \in \mathcal{B}, 1 \le c \le C_\ell\}$. We quantify per-parameter sensitivity under the rectification objective $\mathcal{L}_{\text{rectify}}$ via the empirical (diagonal) Fisher:

$$\widehat{F}_{\theta_i}^{(\text{rect})} = \frac{1}{|\mathcal{D}_{\text{ft}}|} \sum_{(x,y) \in \mathcal{D}_{\text{ft}}} \left\|\nabla_{\theta_i} \mathcal{L}_{\text{rectify}}(x, y)\right\|^2, \qquad \theta_i \in \Theta. \tag{3}$$

For BN affines we set the importance score $s_j := \widehat{F}_{\theta_j}^{(\mathrm{rect})}$ for each $\theta_j \in \Theta_{\mathrm{BN}}$.

*Mask Construction.* Let $K \in \mathbb{N}$ be the pruning budget (optionally $K = \lfloor r |\Theta_{\mathrm{BN}}| \rfloor$ for a ratio $r \in (0,1)$), and let $\tau$ be the $K$-th largest value of $\{s_j : \theta_j \in \Theta_{\mathrm{BN}}\}$. Define the binary mask $M_j \in \{0,1\}$ by

$$M_j = \mathbf{1}\{s_j < \tau\} = \begin{cases} 0, & \text{if } s_j \text{ is among the top-}K \text{ in } \Theta_{\mathrm{BN}}, \\ 1, & \text{otherwise.} \end{cases} \tag{4}$$

**Affine Pruning.** Pruning is one of the most popular methods to remove the effect of a subset of neurons on the model activation and prediction (Li et al., 2021a;c). To remove the backdoor effect, we prune the BatchNorm affine parameters whose corresponding mask values are zero. Concretely, for the $k$-th neuron , we set its weight $w_k = 0$ if $M_k = 0$ and keep it unchanged if $M_k = 1$. Due to the binary masks, pruning is a discrete optimization problem that is difficult to solve within feasible time. To address this, we add a small Gaussian noise to the parameters at the pruned coordinates during fine-tuning. Given the BN affine parameters $\Theta = \{\theta_j\}$ and the affine mask $M$ determined in the previous step, and $\Xi \sim \mathcal{N}(0, \sigma^2 I)$ be i.i.d. noise. We use the masked-and-noised parameters:

$$\theta^u := \tilde{\Theta} = M \odot \Theta + (1 - M) \odot \Xi. \tag{5}$$

**Masked-Gradient Finetuning.** During this process, we zero out the gradient at the affine parameters which are pruned in the previous step. The objective for fine-tuning can be stated as follows:

$$\hat{\theta} := \min_{\theta} \mathbb{E}_{(\boldsymbol{x},y) \in \mathcal{D}_{\mathrm{ft}}} \mathcal{L}_{\mathrm{CE}}(f(\boldsymbol{x}; M \odot \theta^u), y), \qquad \nabla_{\theta} \leftarrow (\nabla_{\theta} \mathcal{L}_{\mathrm{CE}}) \odot M, \tag{6}$$

where $\hat{\theta}$ denotes the current parameters. The mask zeroes gradients only on BN-affine coordinates and leaves all other parameters trainable, preventing drift back toward the backdoored BatchNorm statistics while preserving clean behavior.

## 4 EXPERIMENTS

### 4.1 EXPERIMENTAL SETTINGS

**Attack Setup.** We consider nine distinct backdoor attack strategies: (1) BadNet (Gu et al., 2019), (2) Label Consistent (LC) (Turner et al., 2019), (3) WaNet (Nguyen & Tran, 2021), (4) COMBAT (Huynh et al., 2024), (5) SBL (Pham et al., 2024a), (6) Narcissus Zeng et al. (2023), (7) Refool Liu et al. (2020), (8) Input-aware Nguyen & Tran (2020), and (9) Adaptive Patch Qi et al. (2023). BadNet and LC are representative dirty- and clean-label patch-based attacks, respectively; COMBAT, SBL, and WaNet capture recent optimized and fine-tuning-resilient backdoor designs; Narcissus, Refool, Input-aware, and Adaptive Patch cover more adaptive or semantically driven triggers. We leverage the BackdoorBench (Wu et al., 2022) framework using the authors' provided code for COMBAT and SBL to control trigger pattern, trigger size, and target label. We vary the poisoning rate from 1% to 10%. Unless otherwise stated, we adopt PreAct-ResNet-18 (He et al., 2016) and a 10% fine-tuning ratio by default. We evaluate on three benchmark datasets: CIFAR-10 (Krizhevsky et al., 2009), GTSRB (Stallkamp et al., 2011), and Tiny-ImageNet Le & Yang (2015). Due to space constraints, we report representative results here; additional details and full results are provided in the Appendix.

**Baselines.** We consider nine state-of-the-art defenses covering a range of strategies for mitigating backdoor attacks, from continued training on clean data to model pruning and reinitialization. These defenses include Fine-tuning (FT), NAD (Li et al., 2021a), ANP (Wu & Wang, 2021), FST (Min et al., 2024), TSBD (Lin et al., 2024), I-BAU (Zeng et al., 2021), RNP (Li et al., 2023a), BNP (Zheng et al., 2022b), and UNIT (Cheng et al., 2024). We follow the suggested hyperparameters from BackdoorBench (Wu et al., 2022) and the authors' original codebases.

**Metrics.** Following (Lin et al., 2024; Min et al., 2023; Zhu et al., 2023), we report *C-ACC* (clean accuracy), *ASR* (attack success rate), and *DER* ($\in [0,1]$), which balances ASR reduction against utility: $\mathrm{DER} = \frac{\max(0, \Delta \mathrm{ASR}) - \max(0, \Delta \mathrm{C\text{-}ACC}) + 1}{2}$, where $\Delta \mathrm{ASR}$ and $\Delta \mathrm{ACC}$ are the drop in ASR and C-ACC after applying defense on the backdoored model, respectively. We expect a good defense to have a large C-ACC, DER, and a small ASR. We mark [ASR] when ASR $> 10\%$ . We highlight the best and second best among the nine baselines with **[DER]** and *[DER]* .

Table 1: Comparison of SOTA defenses against multiple backdoor attacks with different fine-tuning ratios on the CIFAR-10 dataset with PreAct-ResNet18.

| Methods | Metrics | FT=0.1 | | | | | | | | FT=0.05 | | | | | | | |
|---|---|---|---|---|---|---|---|---|---|---|---|---|---|---|---|---|---|
| | | BadNet | LC | COMBAT | SBL | Wanet | Refool | Adaptive | Avg. | BadNet | LC | COMBAT | SBL | Wanet | Refool | Adaptive | Avg. |
| Pretrained | C-Acc | 91.44 | 84.19 | 93.94 | 90.52 | 92.67 | 91.65 | 93.08 | 91.07 | 91.36 | 84.51 | 94.13 | 89.76 | 92.90 | 92.00 | 99.33 | 91.99 |
| | ASR | 94.41 | 100.00 | 94.47 | 88.84 | 99.54 | 92.91 | 100.00 | 95.74 | 95.45 | 100.00 | 94.80 | 87.55 | 99.54 | 93.80 | 100.00 | 95.88 |
| | DER | - | - | - | - | - | - | - | - | - | - | - | - | - | - | - | - |
| FT | C-Acc | 90.56 | 90.00 | 93.46 | 90.79 | 92.50 | 91.64 | 92.19 | 91.59 | 88.69 | 90.51 | 94.01 | 89.46 | 92.33 | 91.73 | 92.81 | 91.36 |
| | ASR | 1.47 | 17.53 | 72.83 | 83.85 | 13.91 | 15.54 | 99.94 | 43.58 | 2.22 | 100.00 | 96.17 | 89.92 | 14.97 | 19.21 | 100.00 | 60.36 |
| | DER | 96.03 | 91.24 | 60.58 | 52.50 | 92.73 | 88.68 | 49.59 | 75.90 | 95.28 | 50.00 | 49.94 | 49.85 | 92.00 | 87.16 | 46.74 | 67.28 |
| ANP | C-Acc | 83.51 | 79.17 | 85.18 | 88.77 | 83.62 | 86.92 | 88.33 | 85.07 | 84.40 | 84.51 | 92.14 | 84.48 | 84.83 | 84.19 | 86.91 | 85.92 |
| | ASR | 0.00 | 6.65 | 7.58 | 0.04 | 0.02 | 0.12 | 0.25 | 2.09 | 0.02 | 100.00 | 88.81 | 62.48 | 0.00 | 0.13 | 73.95 | 46.48 |
| | DER | 93.24 | 94.17 | 89.07 | 93.53 | 95.24 | 94.03 | 97.50 | 93.83 | 94.24 | 50.00 | 52.00 | 59.90 | 95.74 | 92.93 | 56.82 | 71.66 |
| NAD | C-Acc | 89.33 | 88.97 | 93.48 | 90.39 | 91.88 | 90.67 | 91.18 | 90.84 | 88.13 | 89.30 | 94.21 | 88.94 | 92.09 | 90.68 | 90.68 | 90.58 |
| | ASR | 2.08 | 18.43 | 70.96 | 64.80 | 9.98 | 9.27 | 90.03 | 37.94 | 2.81 | 59.06 | 97.66 | 73.08 | 1.83 | 6.52 | 65.13 | 43.73 |
| | DER | 95.11 | 90.79 | 61.52 | 61.96 | 94.39 | 91.33 | 54.04 | 78.45 | 94.71 | 70.47 | 50.00 | 56.83 | 98.45 | 92.98 | 63.11 | 75.22 |
| FST | C-Acc | 87.06 | 88.89 | 91.25 | 91.17 | 92.40 | 91.70 | 92.04 | 90.64 | 88.58 | 90.90 | 94.20 | 89.95 | 92.43 | 91.73 | 92.84 | 91.52 |
| | ASR | 2.08 | 2.34 | 30.65 | 0.24 | 0.58 | 3.93 | 0.40 | 5.75 | 1.13 | 0.00 | 90.02 | 30.02 | 0.32 | 5.91 | 41.91 | 24.19 |
| | DER | 93.98 | 98.83 | 80.57 | 94.30 | 99.35 | 94.49 | 99.28 | 94.40 | 95.77 | 100.00 | 52.39 | 78.77 | 99.38 | 93.81 | 75.80 | 85.13 |
| TSBD | C-Acc | 90.13 | 89.06 | 92.91 | 91.43 | 92.48 | 92.24 | 92.40 | 91.52 | 90.00 | 90.74 | 92.28 | 88.14 | 92.43 | 91.85 | 92.12 | 91.08 |
| | ASR | 1.78 | 15.16 | 35.57 | 84.68 | 1.08 | 1.77 | 4.07 | 20.59 | 2.12 | 93.00 | 81.64 | 79.20 | 1.29 | 2.26 | 4.31 | 37.69 |
| | DER | 95.66 | 92.42 | 78.94 | 52.08 | 99.14 | 95.57 | 97.63 | 87.35 | 95.99 | 53.50 | 55.66 | 53.37 | 98.89 | 95.70 | 94.24 | 78.19 |
| I-BAU | C-Acc | 88.13 | 86.33 | 91.01 | 88.20 | 86.52 | 87.87 | 89.84 | 88.27 | 85.71 | 86.21 | 91.85 | 87.61 | 85.77 | 88.17 | 89.83 | 87.88 |
| | ASR | 7.91 | 2.45 | 1.98 | 0.76 | 20.04 | 2.02 | 1.26 | 5.20 | 1.34 | 2.12 | 87.92 | 1.34 | 9.92 | 12.12 | 3.74 | 17.23 |
| | DER | 91.60 | 98.78 | 94.78 | 92.88 | 86.68 | 93.56 | 97.75 | 93.72 | 93.16 | 98.94 | 52.30 | 92.03 | 91.25 | 88.93 | 93.38 | 87.14 |
| BNP | C-Acc | 91.27 | 83.31 | 91.40 | 90.33 | 65.69 | 91.74 | 92.34 | 86.58 | 91.18 | 82.80 | 92.56 | 90.56 | 88.48 | 92.34 | 92.52 | 90.06 |
| | ASR | 13.12 | 0.00 | 24.23 | 90.08 | 47.38 | 3.55 | 9.52 | 26.84 | 16.51 | 0.00 | 13.49 | 93.06 | 14.22 | 40.07 | 72.06 | 35.63 |
| | DER | 90.56 | 99.56 | 83.85 | 49.91 | 62.59 | 94.68 | 94.87 | 82.29 | 89.38 | 99.15 | 89.87 | 50.00 | 90.45 | 76.87 | 60.57 | 79.47 |
| RNP | C-Acc | 87.63 | 80.78 | 92.89 | 87.57 | 90.34 | 54.11 | 90.28 | 83.37 | 84.91 | 82.59 | 93.99 | 72.70 | 86.65 | 51.28 | 89.83 | 80.28 |
| | ASR | 3.76 | 99.93 | 93.09 | 20.57 | 0.17 | 0.00 | 11.67 | 32.74 | 0.07 | 100.00 | 95.39 | 0.01 | 2.47 | 0.00 | 0.87 | 28.40 |
| | DER | 93.42 | 48.33 | 50.17 | 82.66 | 98.52 | 77.69 | 92.77 | 77.65 | 94.47 | 49.04 | 49.93 | 85.24 | 95.41 | 76.54 | 94.82 | 77.92 |
| Unit | C-Acc | 84.66 | 81.36 | 79.70 | 65.64 | 88.05 | 86.75 | 87.57 | 81.96 | 83.30 | 82.07 | 81.04 | 70.15 | 87.19 | 86.84 | 87.09 | 82.53 |
| | ASR | 0.89 | 8.07 | 22.67 | 2.58 | 3.12 | 23.52 | 1.76 | 8.94 | 0.79 | 6.49 | 10.38 | 1.50 | 1.79 | 10.78 | 5.19 | 5.27 |
| | DER | 93.37 | 94.55 | 78.78 | 80.69 | 95.90 | 82.25 | 96.37 | 88.84 | 93.30 | 95.54 | 85.67 | 83.22 | 96.02 | 86.93 | 91.29 | 90.57 |
| Ours | C-Acc | 90.67 | 91.40 | 91.04 | 88.91 | 90.22 | 89.70 | 88.31 | 90.04 | 90.32 | 88.94 | 85.49 | 86.17 | 89.45 | 87.92 | 89.91 | 88.31 |
| | ASR | 1.12 | 2.50 | 10.28 | 2.18 | 4.74 | 1.90 | 3.76 | 3.78 | 4.91 | 5.08 | 7.30 | 4.84 | 2.64 | 3.54 | 1.80 | 4.30 |
| | DER | 98.99 | 98.75 | 93.02 | 97.88 | 96.30 | 94.53 | 95.74 | 96.46 | 96.86 | 97.46 | 91.56 | 95.24 | 96.82 | 93.09 | 94.39 | 95.06 |

## 4.2 MAIN RESULTS

We compare the performance of our method to five other defenses against five representative backdoor attacks. In this section, we present the main results on CIFAR-10 and GTSRB with a 10% poisoning ratio on PreAct-ResNet18 for illustration, which is shown in Table 1 and Table 2.

**Performance of backdoor defenses on CIFAR-10 dataset.** In the CIFAR-10 dataset, our method demonstrates consistent performance across different fine-tuning ratios and outperforms all state-of-the-art defenses on average. At a fine-tuning ratio of 0.1, where the defender can access relatively more clean data, ANP, FST, and our approach are all able to reduce the attack success rate (ASR) while maintaining high clean accuracy (C-ACC). Among them, our method achieves the highest average DER of 95.58%. In contrast, NAD and TSBD already show clear deficiencies against stronger attacks such as COMBAT and SBL, which are either input-dependent or explicitly designed to resist fine-tuning. When the fine-tuning ratio is reduced to 0.05, ANP fails to mitigate several attacks, with ASRs of 100.00% in LC, 88.81% in COMBAT, and 62.48% in SBL, while FST achieves an ASR of 90.06% against COMBAT and 30.02% against SBL. It is worth noting that even some other defenses can maintain slightly higher C-ACC, such as FT and TSBD; these defenses cannot remove the backdoor from the model, which leads to ASR more than 20%. *By comparison, our method continues to maintain the most effective defense against all attacks, achieving an average C-ACC of 90.04% and an average ASR of less than 5%.*

**Performance of backdoor defenses on GTSRB dataset.** On the GTSRB dataset, our method follows a similar trend to that observed on CIFAR-10, consistently outperforming all SOTA defenses and achieving the highest average DER. At a fine-tuning ratio of 0.1, NAD and TSBD again show deficiencies, while ANP and FST also fail to effectively reduce ASR under stronger attacks. ANP achieves a 69.37% ASR and FST achieves 21.65% ASR against COMBAT, illustrating that these approaches struggle when the dataset becomes more complex. The same pattern is evident at a lower fine-tuning ratio of 0.05, where most defenses fail to mitigate at least one attack. *In contrast, our method maintains robust performance across all scenarios, achieving the highest DER of 95.58% for both fine-tuning scenarios.*

Table 2: Comparison of SOTA defenses against multiple backdoor attacks with different fine-tuning ratios on the GTRSB dataset with PreAct-ResNet18.

| Methods | Metrics | FT=0.1 | | | | | | | | FT=0.05 | | | | | | | |
|---|---|---|---|---|---|---|---|---|---|---|---|---|---|---|---|---|---|
| | | BadNet | LC | COMBAT | SBL | Wanet | Refool | Adaptive | Avg. | BadNet | LC | COMBAT | SBL | Wanet | Refool | Adaptive | Avg. |
| Pretrained | C-Acc | 96.85 | 92.45 | 99.07 | 97.36 | 96.19 | 96.11 | 98.65 | 96.67 | 96.94 | 92.41 | 97.97 | 97.29 | 97.45 | 96.74 | 98.76 | 96.79 |
| | ASR | 94.29 | 99.24 | 69.53 | 91.96 | 99.53 | 94.32 | 100 | 92.70 | 94.61 | 99.96 | 76.07 | 90.97 | 99.14 | 92.96 | 100 | 93.39 |
| | DER | - | - | - | - | - | - | - | - | - | - | - | - | - | - | - | - |
| FT | C-Acc | 97.58 | 97.27 | 98.97 | 97.37 | 98.65 | 96.73 | 98.59 | 97.88 | 97.78 | 97.36 | 98.57 | 97.41 | 98.93 | 97.03 | 98.82 | 97.99 |
| | ASR | 67.35 | 97.82 | 65.69 | 90.18 | 34.99 | 87.73 | 100 | 77.68 | 57.86 | 99.06 | 78.18 | 89.27 | 80.17 | 90.16 | 100 | 84.96 |
| | DER | 63.47 | 50.71 | 51.87 | 50.89 | 82.27 | 53.30 | 49.97 | 57.50 | 68.38 | 50.45 | 50.00 | 50.85 | 59.49 | 51.40 | 50.00 | 54.37 |
| ANP | C-Acc | 95.97 | 91.55 | 98.73 | 95.14 | 98.33 | 90.01 | 95.00 | 94.96 | 93.66 | 92.17 | 97.87 | 95.16 | 93.60 | 94.23 | 95.00 | 94.53 |
| | ASR | 13.64 | 1.31 | 69.37 | 1.07 | 0.00 | 0.00 | 0.00 | 12.20 | 0.00 | 35.34 | 54.02 | 0.00 | 0.00 | 0.11 | 0.00 | 12.78 |
| | DER | 89.89 | 98.52 | 49.91 | 94.34 | 99.76 | 94.11 | 98.18 | 89.24 | 95.67 | 99.78 | 60.98 | 94.42 | 99.57 | 95.17 | 99.50 | 89.17 |
| NAD | C-Acc | 97.61 | 97.37 | 99.05 | 97.43 | 98.77 | 96.58 | 98.68 | 97.93 | 97.86 | 96.16 | 98.49 | 97.67 | 99.04 | 96.98 | 98.78 | 97.85 |
| | ASR | 25.53 | 0.37 | 65.23 | 62.59 | 23.60 | 88.39 | 100 | 52.24 | 8.44 | 0.40 | 75.15 | 87.72 | 64.91 | 89.76 | 1.00 | 46.77 |
| | DER | 84.38 | 99.44 | 52.14 | 64.69 | 87.97 | 52.97 | 50.00 | 70.23 | 95.09 | 99.78 | 50.46 | 51.63 | 67.12 | 51.60 | 99.50 | 73.31 |
| FST | C-Acc | 94.50 | 97.28 | 97.71 | 97.22 | 98.89 | 97.88 | 98.45 | 97.42 | 93.64 | 95.17 | 97.81 | 97.33 | 98.66 | 98.04 | 98.27 | 96.99 |
| | ASR | 0.00 | 0.00 | 21.65 | 0.00 | 0.00 | 0.04 | 0.00 | 3.10 | 0.00 | 0.00 | 63.15 | 0.00 | 0.00 | 0.08 | 0.00 | 9.03 |
| | DER | 95.97 | 99.62 | 73.35 | 95.91 | 99.76 | 97.14 | 99.90 | 94.52 | 95.66 | 99.98 | 56.38 | 95.49 | 99.57 | 96.44 | 99.76 | 91.90 |
| TSBD | C-Acc | 98.20 | 97.34 | 99.21 | 97.29 | 94.45 | 98.24 | 98.56 | 97.61 | 97.98 | 96.39 | 98.39 | 96.65 | 86.66 | 97.93 | 85.32 | 94.19 |
| | ASR | 0.00 | 0.16 | 66.58 | 45.42 | 0.21 | 23.42 | 0.57 | 19.48 | 0.03 | 0.80 | 53.99 | 13.22 | 0.00 | 23.74 | 0.00 | 13.11 |
| | DER | 97.15 | 99.29 | 51.48 | 73.24 | 98.79 | 85.45 | 99.67 | 86.44 | 97.29 | 99.58 | 61.04 | 88.56 | 94.18 | 84.61 | 93.28 | 88.36 |
| I-BAU | C-Acc | 92.65 | 95.56 | 99.04 | 93.81 | 98.01 | 96.01 | 96.37 | 95.92 | 93.65 | 95.09 | 96.62 | 93.91 | 97.28 | 96.76 | 96.88 | 95.74 |
| | ASR | 0.14 | 0.77 | 70.31 | 0.78 | 0.22 | 50.69 | 1.55 | 17.78 | 0.03 | 2.50 | 89.97 | 0.81 | 1.68 | 34.55 | 7.55 | 19.58 |
| | DER | 94.98 | 99.24 | 49.99 | 93.82 | 99.66 | 71.77 | 98.09 | 86.79 | 95.65 | 98.73 | 49.33 | 93.39 | 98.65 | 79.21 | 95.29 | 87.18 |
| BNP | C-Acc | 96.73 | 91.78 | 96.46 | 96.88 | 98.48 | 97.36 | 98.54 | 96.60 | 96.42 | 92.11 | 97.13 | 96.11 | 98.42 | 97.88 | 98.57 | 96.66 |
| | ASR | 41.68 | 0.18 | 56.42 | 91.97 | 0.02 | 54.36 | 100 | 49.23 | 65.11 | 0.00 | 73.11 | 91.20 | 0.00 | 67.21 | 100 | 56.66 |
| | DER | 76.25 | 99.20 | 55.25 | 49.76 | 99.76 | 69.98 | 49.95 | 71.45 | 64.49 | 99.83 | 51.06 | 49.41 | 99.57 | 62.88 | 49.91 | 68.16 |
| RNP | C-Acc | 85.48 | 91.48 | 99.07 | 69.54 | 92.88 | 85.90 | 63.32 | 83.95 | 86.43 | 90.33 | 66.06 | 84.77 | 91.94 | 84.84 | 86.77 | 84.45 |
| | ASR | 5.29 | 83.76 | 69.53 | 24.97 | 0.00 | 0.00 | 0.29 | 26.26 | 15.09 | 12.15 | 35.47 | 3.70 | 0.00 | 0.00 | 0.00 | 9.49 |
| | DER | 88.82 | 57.26 | 50.00 | 69.59 | 98.11 | 92.06 | 82.19 | 76.86 | 84.51 | 92.87 | 54.35 | 87.38 | 96.82 | 90.53 | 94.01 | 85.78 |
| Unit | C-Acc | 76.94 | 86.67 | 96.10 | 6.10 | 93.35 | 83.57 | 91.72 | 76.35 | 85.89 | 87.57 | 94.09 | 5.70 | 95.63 | 90.60 | 88.10 | 78.23 |
| | ASR | 16.03 | 2.29 | 6.36 | 99.55 | 0.00 | 52.69 | 36.28 | 30.46 | 0.36 | 0.85 | 28.46 | 100 | 1.17 | 50.81 | 73.59 | 36.46 |
| | DER | 79.18 | 95.59 | 80.10 | 4.37 | 98.35 | 64.55 | 78.40 | 71.50 | 91.60 | 95.59 | 71.87 | 4.21 | 98.08 | 68.01 | 57.88 | 69.82 |
| Ours | C-Acc | 97.43 | 98.22 | 97.38 | 97.10 | 95.19 | 96.75 | 97.93 | 97.14 | 97.86 | 97.40 | 90.63 | 96.27 | 97.16 | 96.09 | 98.13 | 96.22 |
| | ASR | 0.00 | 0.04 | 1.23 | 0.08 | 0.02 | 3.09 | 2.65 | 1.02 | 0.01 | 0.36 | 4.30 | 0.08 | 0.00 | 0.08 | 3.15 | 1.14 |
| | DER | 97.15 | 99.60 | 83.31 | 95.81 | 99.26 | 95.62 | 98.32 | 95.58 | 97.30 | 99.87 | 74.77 | 94.99 | 99.43 | 96.12 | 98.11 | 94.37 |

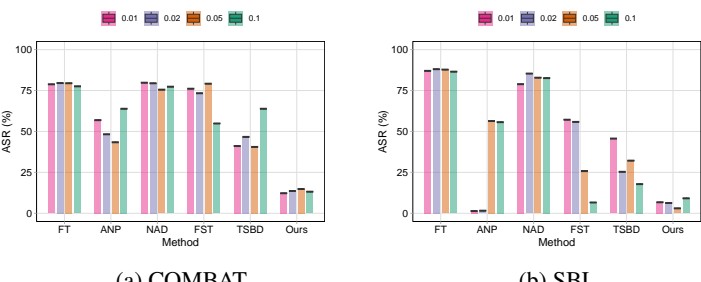

(a) COMBAT        (b) SBL

Figure 4: Defense results (ASR) under various fine-tuning ratio settings with COMBAT and SBL attacks. The experiments are conducted on the CIFAR-10 dataset.

## 4.3 ABLATION STUDIES

In this section, we study the performance of different defenses under varied adversary ability and defender capability. Specifically, we varied the fine-tuning rates from $[0.01, 0.02, 0.05, 0.1]$, where the higher fine-tuning ratio, the more data that the defender can collect to conduct backdoor purification. Then, we vary the poisoning rate to simulate different adversary capability from $[0.01, 0.02, 0.05, 0.1]$. A defense should be stable and effective across the varied settings.

**Effect of fine-tuning ratio.** In this experiment, a larger fine-tuning ratio means a larger amount of data that the defender owns, while a small ratio is considered a more challenging setting. Figure 4 reports ASR (%) for two adaptive backdoors, COMBAT and SBL. The figure shows that the other baselines (FT, NAD, FST, TSBD) are highly sensitive to the fine-tuning ratio: their ASR reduction diminishes as the fine-tuning ratio decreases. Under COMBAT, these defenses still exhibit high ASR even at larger fine-tuning ratios. ANP can suppress ASR at favorable ratios but is unstable at smaller budgets. In contrast, UniBP achieves the lowest ASR across all ratios for both attacks, with the largest gains when the defender can use more than 2% of data for fine-tuning, highlighting superior sample efficiency and stability.

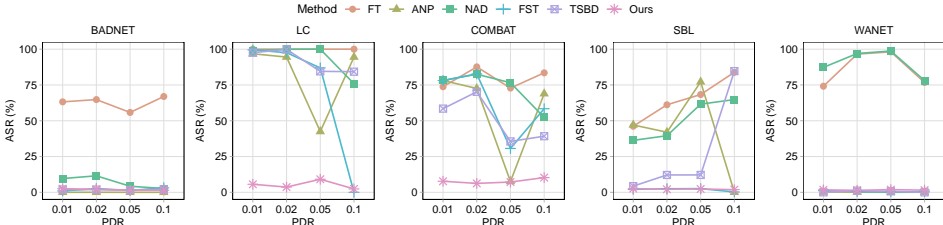

Figure 5: Defense results (ASR) under various poisoned data rate (PDR) settings with LC, COMBAT, and SBL attacks. The experiments are conducted on the CIFAR-10 dataset.

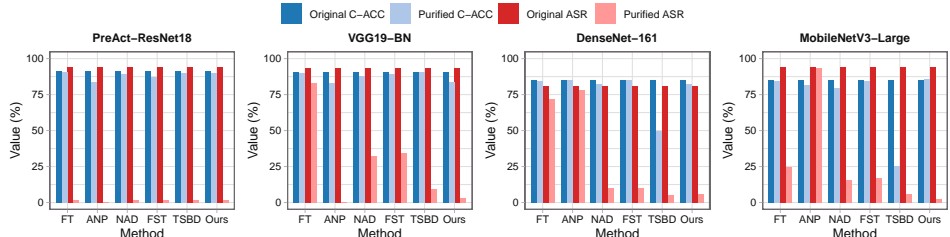

Figure 6: Defense results (ASR) under various model architecture settings with BadNet attack. The experiments are conducted on the CIFAR-10 dataset.

**Effect of data poisoning rate.** Figure 5 shows the effectiveness of all defenses versus different poisoned data rates (PDR) on CIFAR-10 across five attacks (BadNet, LC, Wanet, COMBAT, SBL). From the results, BadNet is the least challenging: all defenses achieve very low ASR. With LC, baselines are more sensitive to different poisoned data rates; several can only reduce ASR to 70% until when PDR is 0.05; whereas UniBP reduces ASR to as low as 0 across all ratios. Under COMBAT and SBL, even more advanced defenses such as TSBD and FST fluctuate widely and often exceed 50% even with larger budgets, and ANP is effective only at selective ratios. In contrast, UniBP is the most effective and stable across all attacks and fine-tuning ratios.

**Analysis on model architecture.** Figure 7 presents the effectiveness of different methods under various backbones: PreAct-ResNet18 (He et al., 2016), VGG19-BN (Simonyan & Zisserman, 2014), DenseNet-161 (Huang et al., 2017), and MobileNetV3-Large (Howard et al., 2019). We report both pre-defense (Original) and post-defense (Purified) clean accuracy (C-ACC) and attack success rate (ASR). Baseline fine-tuning defenses (FT, NAD, FST, TSBD) exhibit pronounced backbone dependence: on VGG19-BN and MobileNetV3-Large, they often leave high purified ASR or incur nontrivial C-ACC drops. ANP can substantially reduce ASR on some backbones (e.g., VGG19-BN) but typically at the cost of noticeable accuracy degradation. In contrast, UniBP consistently achieves the lowest ASR across all four architectures while keeping purified C-ACC close to the original, indicating model-agnostic effectiveness and a better robustness–accuracy trade-off.

## 5 CONCLUSION

We presented UniBP, a universal post-training defense for purifying backdoored models. The approach leverages BatchNorm statistics to expose backdoor footprints, rectifies these statistics on a small clean set, scores BN-affine parameters via a Fisher-based importance measure, prunes the most backdoor-sensitive entries, and fine-tunes with masked gradients—removing trigger pathways without prior knowledge of attack type or location. UniBP consistently attains the lowest ASR while preserving clean accuracy. It is stable across poisoning rates and fine-tuning budgets and operates effectively over a broad mask-ratio range, yielding strong robustness–accuracy trade-offs with modest clean data.

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

# APPENDIX

We conduct all the experiments using PyTorch 2.1.0 (Imambi et al., 2021). All experiments are run on a computer with an Intel Xeon Gold 6330N CPU and an NVIDIA A6000 GPU.

## A    DETAILED EXPERIMENTAL SETUP

### A.1    DATASETS AND PREPROCESSING

**CIFAR-10.** The CIFAR-10 dataset (Krizhevsky et al., 2009) comprises 60,000 32×32 RGB images evenly distributed across 10 classes. We adopt the official split with 50,000 training images and 10,000 test images (6,000 per class in total; 5,000 train and 1,000 test per class). Unless otherwise noted, we follow the standard evaluation protocol on the test set.

**GTSRB (German Traffic Sign Recognition Benchmark).** The GTSRB dataset (Stallkamp et al., 2011) contains 51,839 images across 43 classes, with 39,209 images for training and 12,630 for testing. Following common practice, we use the standard train/test split and resize all images to 32×32 RGB for training and evaluation.

**Tiny-ImageNet (Le & Yang, 2015).** The Tiny-ImageNet dataset is a downscaled subset of ImageNet with 200 classes, each containing 500 training images and 50 validation images, for a total of 100,000 training and 10,000 validation images. All images are 64×64 pixels with RGB channels. Following prior work, we use the official train/validation split and treat the validation set as the test set for evaluation; images are resized to 32×32 when training models that expect CIFAR-style inputs.

### A.2    ATTACK DETAILS

We evaluate our defense against nine SOTA backdoor attacks: BadNets (Gu et al., 2019), LC (Turner et al., 2019), WaNet (Nguyen & Tran, 2021), COMBAT (Huynh et al., 2024), SBL (Pham et al., 2024b), WaNet (Nguyen & Tran, 2021), Narcissus (Zeng et al., 2023), Adaptive-patch (Qi et al., 2023), Input-aware (Nguyen & Tran, 2020) and Refool (Liu et al., 2020). For BadNets, LC, WaNet, Input-aware, Refool we adopt the implementations provided in the BackdoorBench framework and use the default configurations. Since COMBAT, SBL, Narcissus, and Adaptive-Batch are not integrated into BackdoorBench, we incorporated them into our codebase using the official implementations released by the authors[1] [2] [3] [4]. To ensure consistency and comparability across all experiments, we fixed the poisoning ratio at 10%. Examples of poisoned images under each attack is shown in Figure 8.

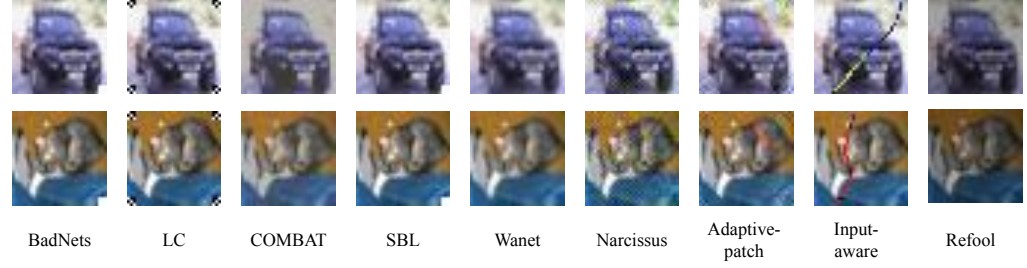

| BadNets | LC | COMBAT | SBL | Wanet | Narcissus | Adaptive-patch | Input-aware | Refool |

Figure 7: Examples of poisoned images on the CIFAR-10 dataset.

### A.3    DEFENSE DETAILS

We compare our method against nine SOTA backdoor defenses: FT, ANP (Wu & Wang, 2021), NAD (Li et al., 2021a), FST (Min et al., 2024), TSBD (Lin et al., 2024), I-BAU (Zeng et al., 2021),

---

[1]https://github.com/VinAIResearch/COMBAT
[2]https://github.com/mail-research/SBL-resilient-backdoors
[3]https://github.com/reds-lab/Narcissus
[4]https://github.com/Unispac/Circumventing-Backdoor-Defenses

RNP (Li et al., 2023a), BNP (Zheng et al., 2022b), and UNIT (Cheng et al., 2024). For FT, ANP, NAD, TSBD, I-BAU, RNP, and BNP, we adopt the implementations and default configurations provided in the BackdoorBench framework. Since FST and Unit are not included in BackdoorBench, we integrated them into our codebase using the publicly released implementation [5] [6]. To ensure fairness across methods, we set the batch size to 256 for all defenses, except for FST, where we follow the original paper and use a batch size of 128. All defenses are trained with a learning rate of 0.002 for 20 epochs. For TSBD, we follow the settings reported in the original paper, fixing the neuron ratio at $n = 0.15$ and the weight ratio at $m = 0.7$. With FST, the coefficient $\alpha$ is set to 0.1. For BNP, we set the target mean shift to $u = 3$, using a search range from $u_{\min} = 0$ to $u_{\max} = 10$ that is discretized into $u_{\text{num}} = 10$ candidate values, and we use a histogram ratio of 0.05. For the alpha-unlearning procedure, we use $\alpha = 0.2$, a clean-classification threshold of 0.80, an unlearning learning rate of 0.01, and a recovering learning rate of 0.1. We train for 20 unlearning epochs and 20 recovering epochs. Neurons are pruned using a threshold-based rule with a maximum pruning ratio of 0.90 and a pruning step size of 0.05. For I-BAU, we follow the original configuration and use $K = 5$ inner optimization steps. For UNIT, we run 300 optimization steps and allow up to 0.03 degradation in clean accuracy.

### A.4 Model architectures and initialization

We evaluate four backbone architectures representative of common vision families:

**PreAct-ResNet-18.** Standard PreAct-ResNet-18; the final classifier is replaced to match the dataset classes (10 for CIFAR-10; 43 for GTSRB).

**VGG19-BN.** VGG19 with batch normalization after each convolutional block; initialized from ImageNet and refit with a dataset-specific classifier.

**DenseNet-161.** ImageNet-pretrained DenseNet-161; the classifier head is replaced to match the target classes.

**MobileNetV3-Large.** ImageNet-pretrained MobileNetV3-Large; the final fully connected layer is replaced to fit the dataset classes.

**Model modifications for purification:** Our purification pipeline interacts primarily with BatchNorm affine parameters and per-channel statistics. We instrument BatchNorm layers to read and optionally reset $\gamma, \beta$ and moving averages ($\mu_{\text{mov}}, \sigma_{\text{mov}}$). For the pruning / affine-mask step we add small, lightweight selection masks per channel (implemented as binary or continuous gates) that can be applied to the BN affine scale term $\gamma$ during inference and finetuning.

### A.5 Hyper-parameters

The pipeline includes separate hyper-parameters for (A) initial training/victim model creation (poisoned model), and (B) purification stages. We list the values used in all experiments unless noted otherwise.

**Training Phase.** Unless otherwise noted, poisoned models are trained using PreAct-ResNet-18 with SGD (momentum 0.9), an initial learning rate of 0.01, weight decay of $5 \times 10^{-4}$, batch size 128, and 100 epochs. The learning rate follows *CosineAnnealingLR*. The random seed is fixed to 0. Unless otherwise specified, standard data augmentation (random horizontal flip and random crop) is applied.

**Fine-tuning Phase.** During fine-tuning, we use SGD with momentum 0.9 and a learning rate in the range $1 \times 10^{-3}$ to $2 \times 10^{-4}$; unless otherwise specified, the training batch size is 128. In the sensitivity-to-fine-tuning-ratio study, we sweep the fine-tuning ratio over $\{1\%, 2\%, 5\%, 10\%\}$ and adjust batch sizes accordingly, i.e., training mini-batch is $\{32, 32, 64, 128\}$, respectively.

Table 3: Tiny-ImageNet results under five backdoor attacks. C-Acc = clean accuracy (%), ASR = attack success rate (%), DER = defense effectiveness ratio (%).

| Attack | Pretrained C-Acc | Pretrained ASR | FT C-Acc | FT ASR | ANP C-Acc | ANP ASR | NAD C-Acc | NAD ASR | FST C-Acc | FST ASR | TSBD C-Acc | TSBD ASR | Ours C-Acc | Ours ASR |
|---|---|---|---|---|---|---|---|---|---|---|---|---|---|---|
| **BadNet** | 47.12 | 94.16 | 55.16 | 90.39 | 47.12 | 94.16 | 49.77 | 29.53 | 26.96 | 0.31 | 52.64 | 54.95 | 48.73 | 17.49 |
| DER | – | | 51.89 | | 50.00 | | 82.32 | | 86.85 | | 69.60 | | **88.34** | |
| **LC** | 56.78 | 67.70 | 56.55 | 67.33 | 54.68 | 59.18 | 56.53 | 70.11 | 28.44 | 0.57 | 54.97 | 16.34 | 50.22 | 2.79 |
| DER | – | | 50.07 | | 53.21 | | 49.87 | | 69.40 | | 74.78 | | **79.18** | |
| **WaNet** | 54.97 | 99.70 | 52.71 | 48.00 | 54.22 | 76.96 | 52.44 | 51.59 | 28.75 | 0.06 | 52.73 | 0.72 | 48.27 | 0.51 |
| DER | – | | 74.72 | | 60.99 | | 72.79 | | 86.71 | | **98.37** | | 96.25 | |
| **Adaptive Patch** | 53.49 | 99.93 | 49.84 | 96.32 | 49.11 | 66.36 | 50.03 | 96.34 | 29.18 | 0.00 | 48.67 | 0.38 | 45.58 | 4.02 |
| DER | – | | 49.98 | | 64.60 | | 50.07 | | 87.81 | | **97.37** | | 94.00 | |
| **Input-aware** | 46.82 | 95.24 | 51.60 | 98.81 | 42.21 | 9.76 | 51.97 | 99.36 | 24.25 | 0.00 | 54.65 | 0.12 | 46.02 | 3.87 |
| DER | – | | 50.00 | | 90.44 | | 50.00 | | 86.34 | | **97.56** | | 95.29 | |
| **AVG-DER** | – | | 55.33 | | 63.85 | | 61.01 | | 83.42 | | 87.54 | | **90.61** | |

# B ADDITIONAL EXPERIMENTAL RESULTS

## B.1 RESULTS WITH TINY-IMAGENET DATASET

We conduct additional experiments on Tiny-ImageNet, which is substantially larger and more complex than CIFAR-10 and GTSRB, and present the results in Table 3. From the results, we can see that UniBP achieves the highest average DER (90.61%) across all five attack types, outperforming TSBD (87.54%) and FST (83.42%). UniBP also consistently suppresses ASR to very low levels across diverse attacks (from standard BadNet to adaptive and input-aware variants) while maintaining clean accuracy around 46-50%. In contrast, FST attains near-zero ASR but drives clean accuracy down to roughly 27%, making it impractical in this setting. Overall, UniBP offers a much better robustness-utility trade-off on Tiny-ImageNet.

## B.2 RESULTS WITH ALL BACKDOOR ATTACKS

Due to space constraints, we only present results with seven attacks in the main paper; here we show all nine attacks in Table 4. From the results, our method consistently achieves the highest average DER across both fine-tuning budgets (93.90 for FT=0.1 and 91.42 for FT=0.05), demonstrating superior robustness to resource constraints. While some baselines perform competitively on specific attacks, where FST achieves 99.35 DER on Wanet and I-BAU reaches 98.78 on LC under FT=0.1, they suffer catastrophic failures on adaptive and clean-label attacks. Notably, traditional fine-tuning (FT) and several specialized defenses (NAD, RNP, BNP) exhibit severe instability against COMBAT and SBL, with DER frequently dropping below 53, indicating near-random performance. These failures stem from violated assumptions: COMBAT optimizes triggers to overlap with target-class features, while SBL explicitly resists fine-tuning through continual learning. In contrast, our method maintains consistently high DER across all nine attacks and both budget settings, with minimal performance degradation under reduced data (only a 2.48 DER drop from FT=0.1 to FT=0.05). This stability across diverse attack families and resource constraints demonstrates the practical effectiveness of our approach in realistic deployment scenarios where attack types are unknown and data is limited.

## B.3 SENSITIVITY TO DATA POISONING RATES

Table 5 presents a comprehensive evaluation of different defense methods (Pretrained, FT, ANP, NAD, FST, TSBD, and UniBP) against a range of backdoor attacks including BadNet, LC, COMBAT, SBL, and Wanet, under varying poisoning data ratios (PDR = 0.1, 0.05, 0.02, 0.01). For each configuration, both model accuracy (MA) and attack success rate (ASR) are reported to highlight the trade-off between maintaining clean accuracy and suppressing malicious behavior. Across the board, baseline Pretrained models show high MA but consistently elevated ASR, indicating vulnerability to all attacks. Fine-tuning (FT) improves resilience to some extent, though it struggles to reduce ASR under

---

[5] https://github.com/AISafety-HKUST/stable_backdoor_purification
[6] https://github.com/Megum1/UNIT

Table 4: Comprehensive evaluation of backdoor defenses across nine attacks under two fine-tuning budgets (FT = 0.1 and FT = 0.05). We report Clean Accuracy (C-Acc), Attack Success Rate (ASR), and Defense Effectiveness Rate (DER). Best results are highlighted in bold.

| Methods | Metrics | BadNet | LC | COMBAT | SBL | Wanet | Narcissus | Adaptive Patch | Input-Aware | Refool | AVG-DER |
|---|---|---|---|---|---|---|---|---|---|---|---|
| **FT = 0.1** | | | | | | | | | | | |
| Pretrained | C-Acc | 91.44 | 84.19 | 93.94 | 90.52 | 92.67 | 93.09 | 93.08 | 90.39 | 91.65 | |
| | ASR | 94.41 | 100.00 | 94.47 | 88.84 | 99.54 | 94.64 | 100.00 | 95.98 | 92.91 | |
| I-BAU | C-Acc | 88.13 | 86.33 | 91.01 | 88.20 | 86.52 | 89.27 | 89.84 | 89.67 | 87.87 | |
| | ASR | 7.91 | 2.45 | 1.98 | 0.76 | 20.04 | 33.01 | 1.26 | 50.90 | 2.02 | |
| | DER | 91.60 | 98.78 | 94.78 | 92.88 | 86.68 | 78.91 | 97.75 | 72.18 | 93.56 | 89.68 |
| UNIT | C-Acc | 84.66 | 81.36 | 79.70 | 65.64 | 88.05 | 87.79 | 87.57 | 80.05 | 86.75 | |
| | ASR | 0.89 | 8.07 | 22.67 | 2.58 | 3.12 | 68.44 | 1.76 | 5.94 | 23.52 | |
| | DER | 93.37 | 94.55 | 78.78 | 80.69 | 95.90 | 60.45 | 96.37 | 89.85 | 82.25 | 85.80 |
| BNP | C-Acc | 91.27 | 83.31 | 91.40 | 90.33 | 65.69 | 93.11 | 92.34 | 89.72 | 91.74 | |
| | ASR | 13.12 | 0.00 | 24.23 | 90.08 | 47.38 | 85.68 | 9.52 | 0.88 | 3.55 | |
| | DER | 90.56 | 99.56 | 83.85 | 49.91 | 62.59 | 54.48 | 94.87 | 97.22 | 94.68 | 80.86 |
| RNP | C-Acc | 87.63 | 80.78 | 92.89 | 87.57 | 90.34 | 92.97 | 90.28 | 86.48 | 54.11 | |
| | ASR | 3.76 | 99.93 | 93.09 | 20.57 | 0.17 | 91.52 | 11.67 | 0.73 | 0.00 | |
| | DER | 93.42 | 48.33 | 50.17 | 82.66 | 98.52 | 51.50 | 92.77 | 95.67 | 77.69 | 76.75 |
| FT | C-Acc | 90.56 | 90.00 | 93.46 | 90.79 | 92.50 | 92.35 | 92.19 | 91.39 | 91.64 | |
| | ASR | 1.47 | 17.53 | 72.83 | 83.85 | 13.91 | 89.81 | 99.94 | 96.50 | 15.54 | |
| | DER | 96.03 | 91.24 | 60.58 | 52.50 | 92.73 | 52.05 | 49.59 | 50.00 | 88.68 | 70.38 |
| ANP | C-Acc | 83.51 | 79.17 | 85.18 | 88.77 | 83.62 | 89.55 | 89.55 | 86.17 | 86.92 | |
| | ASR | 0.00 | 6.65 | 7.58 | 0.04 | 0.02 | 86.77 | 86.77 | 0.16 | 0.12 | |
| | DER | 93.24 | 94.17 | 89.07 | 93.53 | 95.24 | 52.17 | 54.85 | 95.80 | 94.03 | 84.68 |
| NAD | C-Acc | 89.33 | 88.97 | 93.48 | 90.39 | 91.88 | 91.27 | 91.18 | 92.31 | 90.67 | |
| | ASR | 2.08 | 18.43 | 70.96 | 64.80 | 9.98 | 88.06 | 90.03 | 98.80 | 9.27 | |
| | DER | 95.11 | 90.79 | 61.52 | 61.96 | 94.39 | 52.38 | 54.04 | 50.00 | 91.33 | 72.39 |
| FST | C-Acc | 87.06 | 88.89 | 91.25 | 91.17 | 92.40 | 92.18 | 92.04 | 92.67 | 91.70 | |
| | ASR | 2.08 | 2.34 | 30.65 | 0.24 | 0.58 | 93.91 | 0.40 | 0.00 | 3.93 | |
| | DER | 93.98 | 98.83 | 80.57 | 94.30 | 99.35 | 49.91 | 99.28 | 97.99 | 94.49 | 89.86 |
| TSBD | C-Acc | 90.13 | 89.06 | 92.91 | 91.43 | 92.48 | 92.85 | 92.40 | 93.18 | 92.24 | |
| | ASR | 1.78 | 15.16 | 35.57 | 84.68 | 1.08 | 82.16 | 4.07 | 5.43 | 1.77 | |
| | DER | 95.66 | 92.42 | 78.94 | 52.08 | 99.14 | 56.12 | 97.63 | 95.28 | 95.57 | 84.76 |
| **Ours** | C-Acc | 90.67 | 91.40 | 91.04 | 88.91 | 90.22 | 88.37 | 88.31 | 90.61 | 89.70 | |
| | ASR | 1.12 | 2.50 | 10.28 | 2.18 | 4.74 | 14.32 | 3.76 | 5.44 | 1.90 | |
| | DER | 98.99 | 98.75 | 93.02 | 97.88 | 96.3 | 87.80 | 95.74 | 95.27 | 94.53 | 95.36 |
| **FT = 0.05** | | | | | | | | | | | |
| Pretrained | C-Acc | 91.36 | 84.51 | 94.13 | 89.76 | 93.10 | 93.79 | 99.33 | 91.39 | 92.00 | |
| | ASR | 95.45 | 100.00 | 94.80 | 87.55 | 99.88 | 78.05 | 100.00 | 96.50 | 93.80 | |
| I-BAU | C-Acc | 85.71 | 86.21 | 91.85 | 87.61 | 85.77 | 90.02 | 89.83 | 88.54 | 88.17 | |
| | ASR | 3.48 | 2.12 | 87.92 | 1.34 | 9.92 | 71.73 | 3.74 | 62.88 | 12.12 | |
| | DER | 93.16 | 98.94 | 52.30 | 92.03 | 91.25 | 51.27 | 93.38 | 65.39 | 88.92 | 80.74 |
| UNIT | C-Acc | 83.3 | 82.07 | 81.04 | 70.15 | 87.19 | 88.8 | 87.09 | 79.8 | 86.84 | |
| | ASR | 0.79 | 6.49 | 10.38 | 1.5 | 1.79 | 67.23 | 5.19 | 4.32 | 10.78 | |
| | DER | 93.30 | 95.53 | 85.67 | 83.22 | 96.02 | 52.91 | 91.28 | 90.30 | 88.93 | 86.35 |
| BNP | C-Acc | 91.18 | 82.8 | 92.56 | 90.56 | 88.48 | 93.68 | 92.52 | 91.32 | 92.34 | |
| | ASR | 16.51 | 0 | 13.49 | 93.06 | 14.22 | 79.98 | 72.06 | 2.68 | 40.07 | |
| | DER | 89.38 | 99.15 | 89.87 | 50.00 | 90.45 | 49.95 | 60.57 | 96.88 | 76.87 | 78.12 |
| RNP | C-Acc | 84.91 | 82.59 | 93.99 | 72.7 | 86.65 | 92.74 | 89.83 | 64.96 | 51.28 | |
| | ASR | 0.07 | 100 | 95.39 | 0.01 | 2.47 | 73.17 | 0.87 | 0 | 0 | |
| | DER | 94.47 | 49.04 | 49.93 | 85.24 | 95.41 | 51.91 | 94.81 | 85.03 | 76.54 | 75.82 |
| FT | C-Acc | 88.69 | 90.51 | 94.01 | 89.46 | 92.33 | 92.46 | 92.81 | 92.61 | 91.73 | |
| | ASR | 2.22 | 100.00 | 96.17 | 89.92 | 14.97 | 80.92 | 100 | 97.23 | 19.21 | |
| | DER | 95.28 | 0.50 | 49.94 | 49.85 | 92.00 | 49.34 | 46.74 | 50.00 | 87.16 | 57.87 |
| ANP | C-Acc | 84.40 | 84.51 | 92.14 | 84.48 | 84.83 | 86.91 | 86.91 | 89.28 | 84.19 | |
| | ASR | 0.02 | 100.00 | 88.81 | 62.48 | 0.00 | 73.95 | 73.95 | 0.14 | 0.13 | |
| | DER | 94.24 | 0.50 | 52.00 | 59.90 | 95.74 | 48.61 | 56.82 | 97.13 | 92.93 | 66.43 |
| NAD | C-Acc | 88.13 | 89.30 | 94.21 | 88.94 | 92.09 | 91.35 | 90.68 | 92.93 | 90.68 | |
| | ASR | 2.81 | 59.06 | 97.66 | 73.08 | 1.83 | 75.96 | 65.13 | 83.74 | 6.52 | |
| | DER | 94.71 | 70.47 | 50.00 | 56.83 | 98.45 | 49.83 | 63.11 | 56.38 | 92.98 | 70.31 |
| FST | C-Acc | 88.58 | 90.90 | 94.20 | 89.95 | 92.43 | 92.84 | 92.84 | 93.32 | 91.73 | |
| | ASR | 1.13 | 0.00 | 90.02 | 30.02 | 0.32 | 81.2 | 41.91 | 0.01 | 5.91 | |
| | DER | 95.77 | 100.00 | 52.39 | 78.77 | 99.38 | 49.53 | 75.80 | 98.25 | 93.81 | 82.63 |
| TSBD | C-Acc | 90.00 | 90.74 | 92.28 | 88.14 | 92.43 | 90.58 | 92.12 | 93.31 | 91.85 | |
| | ASR | 2.12 | 93.00 | 81.64 | 79.20 | 1.29 | 74.11 | 4.31 | 1.69 | 2.26 | |
| | DER | 95.99 | 53.50 | 55.66 | 53.37 | 98.89 | 50.37 | 94.24 | 97.41 | 95.70 | 77.24 |
| **Ours** | C-Acc | 90.32 | 88.94 | 84.42 | 81.28 | 89.45 | 85.17 | 89.91 | 87.30 | 87.92 | |
| | ASR | 4.91 | 5.08 | 13.63 | 4.13 | 2.64 | 9.87 | 1.80 | 5.57 | 3.54 | |
| | DER | 94.75 | 97.46 | 85.73 | 87.47 | 96.73 | 79.78 | 94.39 | 93.42 | 93.09 | 91.42 |

Table 5: Performance comparison of different defense methods (Pretrained, FT, ANP, NAD, FST, TSBD, and PBP) against multiple backdoor attacks (BadNet, LC, COMBAT, SBL, and Wanet) under varying poisoning data ratios (PDR = 0.1, 0.05, 0.02, 0.01). The table reports the model accuracy (MA) and attack success rate (ASR) in percentage.

| Attacks | PDR | Pretrained | | FT | | ANP | | NAD | | FST | | TSBD | | Ours | |
|---|---|---|---|---|---|---|---|---|---|---|---|---|---|---|---|
| | | C-ACC | ASR | C-ACC | ASR | C-ACC | ASR | C-ACC | ASR | C-ACC | ASR | C-ACC | ASR | C-ACC | ASR |
| BadNet | 0.1 | 91.44 | 94.41 | 91.01 | 66.94 | 84.10 | 0.00 | 89.62 | 2.66 | 91.48 | 3.13 | 91.67 | 2.10 | 89.82 | 1.47 |
| | 0.05 | 92.15 | 90.30 | 91.37 | 55.80 | 85.05 | 0.00 | 90.58 | 4.24 | 91.37 | 1.30 | 92.19 | 1.43 | 87.85 | 1.50 |
| | 0.02 | 92.81 | 81.47 | 91.77 | 64.78 | 86.49 | 0.01 | 91.34 | 11.52 | 92.33 | 2.58 | 92.18 | 1.88 | 87.59 | 2.09 |
| | 0.01 | 93.34 | 71.21 | 92.39 | 63.20 | 85.05 | 0.01 | 91.12 | 9.63 | 92.50 | 0.94 | 93.02 | 1.94 | 87.68 | 2.51 |
| LC | 0.1 | 84.19 | 100.00 | 92.80 | 100.00 | 84.67 | 94.48 | 91.39 | 75.88 | 91.08 | 0.00 | 91.08 | 84.27 | 89.09 | 2.36 |
| | 0.05 | 93.32 | 100.00 | 92.26 | 100.00 | 91.55 | 42.47 | 90.82 | 100.00 | 92.22 | 86.91 | 91.66 | 84.52 | 86.19 | 9.13 |
| | 0.02 | 93.39 | 100.00 | 92.68 | 100.00 | 84.67 | 94.48 | 91.57 | 99.91 | 92.38 | 97.55 | 92.98 | 99.84 | 83.06 | 3.67 |
| | 0.01 | 93.54 | 99.97 | 92.36 | 100.00 | 88.77 | 96.80 | 91.51 | 99.01 | 92.59 | 99.52 | 92.78 | 97.54 | 83.04 | 5.61 |
| COMBAT | 0.1 | 85.05 | 99.23 | 91.90 | 83.47 | 84.20 | 68.90 | 91.80 | 52.28 | 92.27 | 58.40 | 92.56 | 39.20 | 91.04 | 10.28 |
| | 0.05 | 93.94 | 94.47 | 93.46 | 72.83 | 85.18 | 7.58 | 98.41 | 76.56 | 91.25 | 30.65 | 92.91 | 35.57 | 87.90 | 7.18 |
| | 0.02 | 93.90 | 85.04 | 94.20 | 87.63 | 85.05 | 72.56 | 93.61 | 82.51 | 93.90 | 82.73 | 92.76 | 70.21 | 89.72 | 6.23 |
| | 0.01 | 94.14 | 83.67 | 93.49 | 73.76 | 93.40 | 78.12 | 93.40 | 78.12 | 93.60 | 78.07 | 92.78 | 58.46 | 87.29 | 7.72 |
| SBL | 0.1 | 90.52 | 88.84 | 90.79 | 83.85 | 88.77 | 0.04 | 90.39 | 64.80 | 91.17 | 0.24 | 91.43 | 84.68 | 83.25 | 1.86 |
| | 0.05 | 90.02 | 79.35 | 89.94 | 68.31 | 85.53 | 77.16 | 89.68 | 61.60 | 90.06 | 2.58 | 89.55 | 12.15 | 87.81 | 2.33 |
| | 0.02 | 90.25 | 68.27 | 90.33 | 61.16 | 87.59 | 41.98 | 90.03 | 39.53 | 90.17 | 2.48 | 90.25 | 12.12 | 88.04 | 2.13 |
| | 0.01 | 90.50 | 47.07 | 90.39 | 46.14 | 90.50 | 47.07 | 89.98 | 36.30 | 90.49 | 2.08 | 91.01 | 4.33 | 88.44 | 2.60 |
| Wanet | 0.1 | 93.40 | 99.97 | 93.75 | 76.73 | 84.46 | 0.08 | 93.72 | 78.07 | 93.04 | 0.30 | 77.64 | 0.00 | 90.22 | 1.52 |
| | 0.05 | 93.37 | 99.87 | 93.89 | 98.14 | 85.33 | 0.01 | 93.88 | 98.73 | 93.32 | 0.27 | 70.42 | 0.70 | 90.09 | 1.88 |
| | 0.02 | 93.43 | 99.38 | 93.54 | 96.54 | 88.44 | 0.13 | 93.77 | 96.98 | 93.37 | 0.50 | 43.97 | 1.15 | 89.47 | 1.39 |
| | 0.01 | 93.81 | 98.48 | 93.80 | 74.12 | 85.88 | 0.21 | 93.83 | 87.26 | 93.18 | 1.13 | 75.40 | 0.00 | 89.71 | 1.73 |

low PDRs. ANP and NAD demonstrate stronger backdoor mitigation, often reducing ASR close to zero, but at the cost of a slight drop in MA in some cases. FST and TSBD provide a more balanced trade-off, achieving high MA while substantially lowering ASR in multiple attack settings. Notably, `UniBP` consistently achieves competitive MA while keeping ASR at very low levels, especially under LC and COMBAT attacks, showcasing its robustness under challenging conditions. Overall, the results emphasize that while most defenses reduce ASR to some degree, methods like NAD, FST, TSBD, and particularly `UniBP` stand out in delivering both strong protection and reliable utility.

## B.4 SENSITIVITY TO MODEL ARCHITECTURES

### B.4.1 MODEL ARCHITECTURES WITH BATCHNORM LAYERS

Table 6: Comparison of defenses against BadNet across models (original vs. purified). Values are clean accuracy (C-ACC) and attack success rate (ASR), both in %.

| Model | Tag | FT | | ANP | | NAD | | FST | | TSBD | | Ours | |
|---|---|---|---|---|---|---|---|---|---|---|---|---|---|---|
| | | C-ACC | ASR | C-ACC | ASR | C-ACC | ASR | C-ACC | ASR | C-ACC | ASR | C-ACC | ASR |
| VGG19_BN | Original | 90.84 | 93.41 | 90.84 | 93.41 | 90.84 | 93.41 | 90.84 | 93.41 | 90.84 | 93.41 | 90.84 | 93.41 |
| | Purified | 89.72 | 82.77 | 83.38 | 0.00 | 88.09 | 32.15 | 89.05 | 34.74 | 90.67 | 9.38 | 83.48 | 2.84 |
| DenseNet161 | Original | 84.97 | 80.91 | 84.97 | 80.91 | 84.97 | 80.91 | 84.97 | 80.91 | 84.97 | 80.91 | 84.97 | 80.91 |
| | Purified | 84.58 | 71.77 | 85.28 | 78.18 | 82.51 | 10.30 | 84.87 | 10.13 | 49.72 | 4.88 | 82.03 | 5.70 |
| MobileNetV3-Large | Original | 85.12 | 94.12 | 85.12 | 94.12 | 85.12 | 94.12 | 85.12 | 94.12 | 85.12 | 94.12 | 85.12 | 94.12 |
| | Purified | 84.24 | 24.71 | 81.74 | 93.36 | 79.80 | 15.62 | 84.20 | 17.15 | 25.13 | 5.86 | 85.78 | 2.23 |
| PreAct-ResNet18 | Original | 91.44 | 94.41 | 91.44 | 94.41 | 91.44 | 94.41 | 91.44 | 94.41 | 91.44 | 94.41 | 91.44 | 94.41 |
| | Purified | 90.56 | 1.47 | 83.51 | 0.00 | 89.33 | 2.08 | 87.06 | 2.08 | 90.13 | 1.78 | 89.82 | 1.47 |

Table 6 compares Original vs. Purified C-ACC/ASR under BadNet across four backbones. Baseline fine-tuning defenses (FT, NAD, FST, TSBD) show pronounced backbone dependence: on VGG19-BN and DenseNet161 they often leave high purified ASR ($\sim 10\%$), and on DenseNet161 and MobileNetV3-Large, TSBD substantially reduces C-ACC. ANP lowers ASR on some backbones (VGG19-BN, PreAct-ResNet18) but with noticeable accuracy drops (7-8%) and fails on the others (ASR remains high, often above 70-90%). In contrast, `UniBP` keeps ASR low across all architectures (about 1–6%) while maintaining purified C-ACC close to the original (typically within a few points), indicating backbone-agnostic effectiveness and a better robustness–accuracy trade-off.

### B.4.2 MODEL ARCHITECTURES WITHOUT BATCHNORM LAYERS

We further investigate an adapted version of our method for non-BN models such as Vision Transformers by recognizing that ViTs use LayerNorm instead of BatchNorm and follow a Pre-LN

architecture where normalization precedes computation (LN → MLP) rather than following it (Conv → BN). Since LayerNorm does not maintain running statistics like BatchNorm, we manually collect reference statistics by hooking LayerNorm inputs during forward passes on clean data, computing mean and variance across feature dimensions over multiple batches. Our key architectural insight is that in Pre-LN Transformers, the MLP feed-forward blocks between consecutive LayerNorm layers directly determine the input distribution to the subsequent normalization—therefore, we strategically target only MLP parameters (fc1, fc2 weights/biases) for FIM computation while explicitly excluding LayerNorm parameters themselves through `_is_mlp_param()` filtering. We attach hooks to LayerNorm layers to capture their input statistics and compute an alignment loss $\mathcal{L} = \|\mu_{\text{input}} - \mu_{\text{ref}}\|^2 + \lambda\|\sigma_{\text{input}} - \sigma_{\text{ref}}\|^2$ against a reinitialized clean baseline model, which serves as our reference for "normal training dynamics." When this loss backpropagates, gradients flow through the LayerNorm back to the upstream MLP blocks, and the accumulated squared gradients (FIM scores) reveal which MLP parameters are most critical to producing backdoor-specific activation distributions—parameters with high FIM resist alignment with clean statistics because they were optimized on poisoned data. Finally, we prune only the top-ranked MLP parameters via noise injection, preserving the normalization layers while disrupting the backdoor pathway hidden in the feed-forward sublayers where Transformer backdoors typically reside.

Table 7: Performance of UniBP on Vision Transformer models with different backdoor attacks and poisoning rates (FT).

| Metric | BadNet | | LC | |
|---|---|---|---|---|
| | FT=0.1 | FT=0.05 | FT=0.1 | FT=0.05 |
| C-Acc (Pretrained) | 0.9286 | 0.9062 | 0.8736 | 0.8731 |
| ASR (Pretrained) | 0.9542 | 0.9302 | 1.0000 | 1.0000 |
| C-Acc (Ours) | 0.9325 | 0.9310 | 0.9529 | 0.9467 |
| ASR (Ours) | 0.0034 | 0.0101 | 0.0230 | 0.0006 |
| DER (Ours) | 0.9754 | 0.9601 | 0.9885 | 0.9997 |

As shown in Table 7, our adapted method effectively eliminates backdoors across all attack scenarios (ASR reduced to near-zero) while preserving or even improving clean accuracy, demonstrating that targeting MLP parameters via LayerNorm statistics successfully disrupts backdoor pathways without degrading model performance.

## B.5 ABLATION STUDY

We sweep the mask ratio $K$, the primary control in our method, and summarize the outcomes in Figure 8. Across all settings, C-ACC decreases smoothly as $K$ increases, with only a small drop (typically $\leq 5$ points) inside the shaded range and a sharp decline once $K \geq 0.10 \times 10^{-3}$. ASR remains low overall, generally within 1–5%; LC at 10% poisoning shows a mild bump near $K \approx 0.06 \times 10^{-3}$, but the trend is otherwise flat. Increasing $K$ beyond the shaded range yields little additional ASR reduction while causing substantial loss in clean accuracy, most notably for BadNet at 5% poisoning. Small pruning budgets within the highlighted range therefore, provide the best trade-off, keeping ASR low with minimal impact on clean performance across both attack families and poisoning rates.

## B.6 ADDITIONAL PLOTS

Figure 9 summarizes how different backdoor families distort the representation space and BatchNorm statistics. The t-SNE plots (top) show that BadNet and LC largely blend poisoned samples into the target-class manifold, yielding only mild geometric separation; WANET induces a moderate shift with partially segregated clusters; SBL creates a compact, outlying poisoned cluster that is clearly detached from clean structure; COMBAT, which mixes patch- and distributional cues, produces overlap similar to BadNet but with denser target-class concentration. The histograms of BN per-channel means (bottom) mirror these trends: BadNet and LC exhibit near-overlapping clean vs. backdoored distributions (small mean shifts), WANET shows a visible but modest shift, and SBL displays a pronounced displacement of the backdoored distribution. COMBAT lies between these extremes. Overall, attacks that strongly perturb intermediate distributions (e.g., SBL) leave a larger BN footprint, whereas patch-like attacks (BadNet/LC) are more stealthy in BN space—motivating a

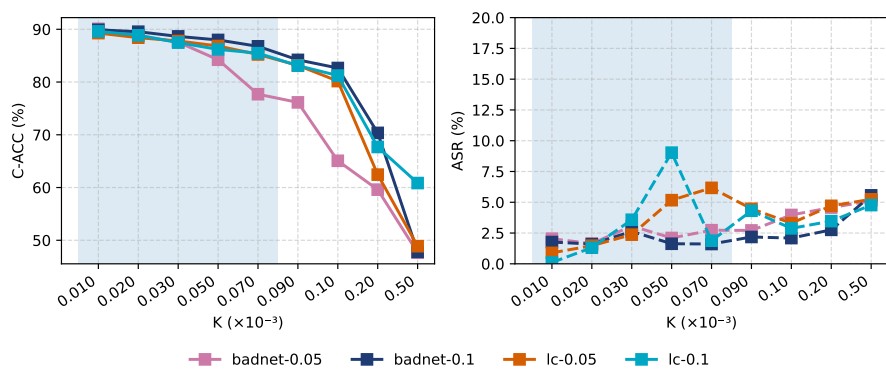

Figure 8: Effect of pruning budget $K$ on clean accuracy (C-ACC, left) and attack success rate (ASR, right) under BadNet and LC with poisoning rates 5% and 10% on CIFAR-10. The shaded band marks the stable operating range ($K \in [0.010, 0.070] \times 10^{-3}$).

rectification objective that leverages BN statistics while also requiring parameter-level masking to handle the subtler cases. *Though these attacks are different in manner and how the trigger is crafted, the shift phenomenon in BN statistics could be leveraged to defend against these attacks.*

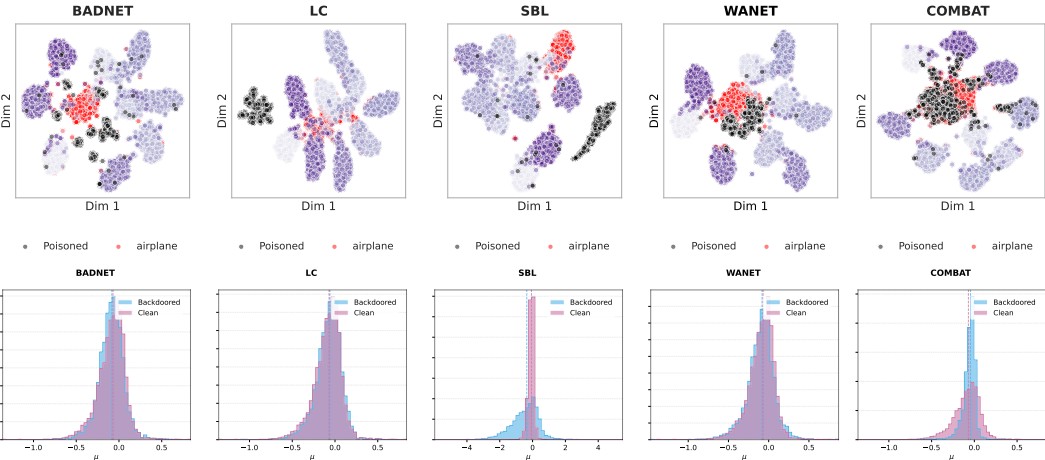

Figure 9: t-SNE of feature embeddings of different attack strategies and their effect on BN layers' statistic CIFAR-10 of different attack families.

## C  ANALYSIS AND DISCUSSION

### C.1  COMPUTATIONAL OVERHEAD

We compare the computational cost of different defenses by measuring their running time on 5000 CIFAR-10 images with a PreAct-ResNet-18 backbone, and present the results in Table 8 and Figure 10. From the results, we can see that our method achieves the highest average DER (93.90%) among all evaluated defenses while maintaining a moderate running time of 228 seconds. In particular, it is substantially faster than TSBD (1529s) and ANP (414s), and remains in the same ballpark as lighter baselines such as NAD (129s) and BNP (153s). Several methods with comparable or lower DER (e.g., FT, ANP, UNIT) require considerably more computation, indicating that our approach offers a more favorable robustness–efficiency trade-off. Overall, these results suggest that our defense is not only effective but also computationally practical for deployment in realistic FL settings.

Table 8: Running time and average DER of different defenses on 5000 CIFAR-10 images using PreAct-ResNet-18 under the same hardware setting.

| Metric | FT | ANP | NAD | FST | TSBD | BNP | I-BAU | RNP | UNIT | Ours |
|---|---|---|---|---|---|---|---|---|---|---|
| Running Time (s) | 95 | 414 | 129 | 157 | 1529 | 153 | 132 | 181 | 421 | 228 |
| Avg. DER | 70.38 | 89.42 | 72.39 | 89.86 | 84.76 | 80.86 | 89.68 | 76.75 | 85.80 | **93.90** |

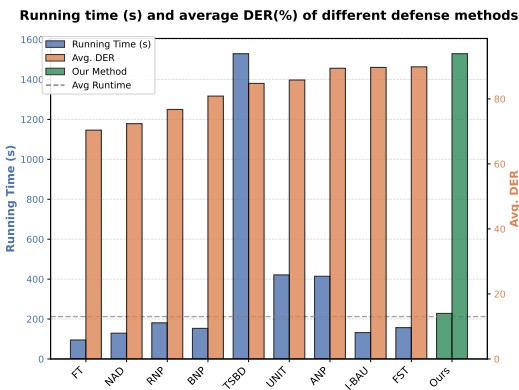

Figure 10: Running time vs. average DER for different defenses on CIFAR-10

## C.2 ADAPTIVE ATTACKS

We evaluate robustness against a **strong adaptive adversary** who has complete knowledge of our defense mechanism and explicitly attempts to evade detection by preserving benign batch normalization statistics. The attacker augments the standard backdoor poisoning objective with a regularization term that penalizes deviations from clean BN statistics across all layers. Formally, assuming access to reference statistics $\mu_\ell^c$ and $v_\ell^c$ from benign data, the adaptive attack minimizes:

$$\mathcal{L}_{\text{adaptive}} = \mathbb{E}_{(\mathbf{x},y)\sim\mathcal{D}}\left[\mathrm{D}_{\mathrm{CE}}(y, f(\delta(\mathbf{x})))\right] + \gamma \sum_{\ell=1}^{L} \mathbb{E}_{\mathbf{x}\sim\mathcal{X}}\left[\|\hat{\mu}_\ell - \mu_\ell^c\|_2 + \lambda\|\hat{v}_\ell - v_\ell^c\|_2\right],$$

where the first term ensures high attack success rate and the second term explicitly aligns the backdoored model's BN statistics with those of a clean model. We systematically evaluate this adaptive attack across regularization strengths spanning five orders of magnitude ($\gamma \in \{0, 0.01, 0.1, 1.0, 10.0, 100.0\}$). Table 9 shows that across all viable settings, our defense maintains ASR below 9% while the pretrained backdoored model exhibits ASR above 94%, demonstrating that UniBP remains highly effective even when attackers explicitly target the BN-based detection mechanism. This robustness stems from a fundamental tension: backdoor functionality inherently requires trigger-dependent feature representations that create distributional shifts detectable in BN statistics, and suppressing these shifts to evade detection directly undermines the attack's effectiveness.

## C.3 MITIGATING THE CLEAN ACCURACY TRADE-OFF

We acknowledge that UniBP may incur a slightly larger clean-accuracy drop compared to some baselines. However, we view this as an inherent and well-documented trade-off in pruning-based defenses operating under zero-adversary-knowledge assumptions: any method that aggressively suppresses backdoor-related capacity without access to the true trigger or strong side information will inevitably sacrifice some clean performance, as observed in prior work such as ANP and NAD. Critically, UniBP is the only defense effective against all tested backdoor attacks, including challenging sample-specific and adaptive variants where other methods fail to provide adequate protection. In contrast, methods that preserve marginally higher clean accuracy often leave non-trivial residual backdoor risk, making the comparison fundamentally asymmetric. We further demonstrate

Table 9: Performance under adaptive attacks with BN-alignment regularization across varying regularization strengths $\gamma$.

| Method | Metric | $\gamma = 0$ | $\gamma = 0.01$ | $\gamma = 0.1$ | $\gamma = 1.0$ | $\gamma = 10.0$ | $\gamma = 100.0$ |
|--------|--------|------|---------|--------|--------|---------|----------|
| Pretrained | ACC | 91.44 | 91.18 | 89.34 | 89.06 | 88.78 | — NaN |
| Pretrained | ASR | 94.41 | 94.20 | 96.24 | 95.62 | 95.41 | — NaN |
| Ours | ACC | 89.82 | 90.55 | 87.06 | 87.22 | 86.67 | — NaN |
| Ours | ASR | 1.47 | 2.39 | 3.66 | 8.48 | 3.01 | — NaN |

Table 10: Clean accuracy recovery with minimal additional fine-tuning data. Adding a small fraction $r\%$ of extra clean data after UniBP fully recovers accuracy while maintaining strong backdoor suppression.

| Additional Data Ratio | BadNet | | | LC | | |
|-----------------------|--------|-----|-----|--------|--------|-----|
| | ACC | ASR | DER | ACC | ASR | DER |
| Pretrained | 91.44 | 94.41 | — | 84.19 | 100.00 | — |
| r=0.00 | 89.82 | 1.47 | 95.66 | 89.09 | 2.36 | 98.82 |
| r=0.01 | 92.03 | 0.92 | 96.75 | 92.62 | 0.07 | 99.97 |
| r=0.02 | 91.84 | 1.12 | 96.65 | 92.61 | 0.07 | 99.97 |
| r=0.05 | 91.60 | 1.00 | 96.71 | 92.75 | 0.04 | 99.98 |

that this trade-off is mitigable rather than fundamental. We conducted an ablation study where an additional r% of clean training data is used for a third fine-tuning step after UniBP completes its pruning and recovery phases. As shown in Table 10, adding even a small fraction of additional clean data is sufficient to recover—or even exceed fully—the pretrained model's clean accuracy, while maintaining near-zero ASR and near-perfect backdoor removal. Notably, using just 1% additional data improves accuracy by over 2% on BadNet (from 89.82% to 92.03%) and 3.5% on LC (from 89.09% to 92.62%), surpassing the original pretrained accuracy in both cases while keeping ASR below 1%, making our method achieve comparable clean accuracy with other baselines. The improvements plateau beyond this point, with marginal gains at higher data ratios, suggesting that minimal additional resources are needed for effective mitigation. These results demonstrate that in practical deployment scenarios, practitioners can achieve a favorable balance between robustness and utility with modest extra cost, while maintaining the defense's core advantage of comprehensive protection against diverse backdoor threats.

## D  LIMITATIONS

We note several limitations that contextualize our results and suggest directions for future work. First, the method assumes access to a small hold-out clean set to estimate BatchNorm statistics and to drive affine-mask learning; its size, class coverage, and label quality materially affect stability and final accuracy. In extremely low-data or noisy-label regimes, the rectification signal can weaken, and the fully unsupervised setting (no clean data) is outside our scope. Second, while we evaluate adaptive variants, a stronger adversary that co-designs triggers to survive BN-affine reset and pruning, perturbs or hijacks running statistics during poisoning, or disperses triggers to reduce gradient salience could diminish effectiveness; developing defenses with explicit guarantees against such adaptive strategies remains open. Third, our study focuses on image classification with BN-based architectures; extending the approach to other modalities (e.g., audio, NLP) or tasks (e.g., detection, segmentation), and to models using alternative normalizations (e.g., LayerNorm, GroupNorm), will require adapting both the rectification objective and the mask parameterization. UniBP currently assumes access to a small clean subset, which is a common setup in recent defenses (e.g., I-BAU, RNP, ANP). To relax this assumption, combining UniBP with data-free techniques is a viable direction. For instance, one could employ generative models (e.g., GANs or diffusion models) trained on benign data to approximate clean samples and recover BN statistics. The main challenge lies in ensuring

that generated samples faithfully preserve the statistical structure of the original training data. We acknowledge this as an exciting area for future work and will add it to the discussion.

## E  BROADER IMPACT

**Positive impacts.** The method strengthens deployed classifiers against poisoning/backdoor threats, improving robustness in safety-critical settings (e.g., automotive perception, medical imaging).

**Dual use.** Defensive techniques can inform stronger, defense-aware attacks. We will release code with clear usage guidance and a responsible license, and provide deployment recommendations (e.g., separate clean validation, periodic re-evaluation), limiting exploit-ready details to what is necessary for reproducibility.

**Privacy.** The approach assumes a small clean dataset; when data are sensitive, practitioners should minimize collection, de-identify inputs, restrict access, and follow IRB requirements.

**Responsible disclosure.** We support coordinated disclosure to affected stakeholders and commit to sharing only information needed for verification and remediation.

