# OpenReview forum: "UniBP: Toward Universal Backdoor Purification via Fine-Tuning"
_ICLR.cc/2026/Conference — Submitted to ICLR 2026_

### Official Review · Reviewer_nswQ · 2025-10-21

**Soundness:** 2
**Presentation:** 2
**Contribution:** 2
**Rating:** 2
**Confidence:** 5

**Summary:**

The paper adjusts the parameters of batch normalization layers to mitigate backdoor poisoning.

**Strengths:**

Backdoors pose a challenging problem.

**Weaknesses:**

The following uncited paper adjusts batch-normalization parameters
for backdoor defense.  So, it should have been cited and
compared against.
[1] X. Li et al.  Backdoor Mitigation by Correcting the Distribution of Neural Activations. Elsevier Neurocomputing  614, 21 January 2025. http://arxiv.org/abs/2308.09850

Re. line 145: Though some papers do assume a strong adversary (insider) who controls the training _process_, typically backdoor poisoning can be effectively accomplished by just inserting poisoned examples into the training dataset.

Re. line 148,149: The statement is odd because a very large
number of prior papers on backdoor defense, particularly
inversion/reverse-engineering approaches,  make exactly this
"post training" assumption.

Is the (cited) I-BAU method compared against in Table 1?

The fonts in the figures and tables are too small.

**Questions:**

See the above.

---

> ### Author Response · Authors · 2025-11-26
> **Response to Reviewer#nswQ (1)**
>
> We would like to thank the reviewer for the constructive and thoughtful feedback. We appreciate the time and effort spent evaluating our work, and we address each concern in detail as follows.
>
> **W1. Uncited paper BNA**
>
> Thanks for pointing this out. We will add the BNA paper [1] to the related work section and discuss a theoretical comparison.
>
> The key differences between BNA and our method are:
>
> - BNA explicitly relies on the distributional shift between clean and triggered data at each neuron and then minimizes the KL divergence between these two distributions.
> - However, this setting violates our threat model and makes a direct comparison unfair: BNA requires an estimated trigger function (Algorithm 1 in the BNA paper), which is not available in our scenario and is not straightforward to obtain in practice.
> - In contrast, our method does **not** require any triggered dataset or explicit trigger estimation. UniBP only observes how BN-layer parameters behave when a freshly reinitialized model is aligned toward the BN statistics of a backdoored model, making it more practical and less dependent on strong assumptions about the attacker’s trigger.
>
> **W2. Strong adversary (insider) assumptions**
> We appreciate this comment and would like to clarify our threat model as follows.
> While many backdoor attacks can be accomplished through data poisoning alone, we explicitly consider stronger adversarial scenarios (such as the insider threat in Pham et al. SBL [2]) to demonstrate the robustness of our defense. We emphasize that our paper shows UniBP is effective against both standard data poisoning attacks and these more extreme insider threat scenarios.
> We believe considering stronger threat models actually strengthens our contribution for two reasons:
>
> (1) If a defense can effectively handle extreme scenarios with insider adversaries who control the training process, it provides stronger guarantees for the more common case of data poisoning attacks.
> (2) As noted by reviewers R#uAhn and R#Yqvj, effective backdoor defenses must consider adaptive adversaries who are aware of the defense mechanism and can attempt to bypass it. Evaluating against stronger threat models helps demonstrate resilience against such adaptive strategies.
>
> Therefore, rather than limiting our contribution, *this threat model assumption strengthens the broader applicability and robustness of UniBP* across a spectrum of attack scenarios, from standard data poisoning to sophisticated insider threats.
>
> **W3. “Post-training” assumption**
>
> We agree that the post-training setting is not new and has been widely adopted in prior backdoor defenses (e.g., I-BAU, FST, ANP, RNP). Our intention was not to claim that this assumption is unusual, but to clarify the threat model we focus on. We will revise the wording accordingly.
>
> We argue that the post-training (model-only) regime is both practical and important for the following reasons: (i) in many real-world scenarios, users obtain a pre-trained model from an external provider or model zoo and have no control over, or access to, the original training pipeline; (ii) retraining from scratch on large-scale data is often prohibitively expensive or infeasible; and (iii) several strong baselines already operate under this assumption, which makes it a natural setting for fair comparison.
>
> Our contribution is therefore not the assumption itself, but a defense that remains effective in this realistic, model-only post-training scenario, without requiring access to the original training data or trigger information.

---

> ### Author Response · Authors · 2025-11-26
> **Response to Reviewer#nswQ (2)**
>
> **W4. Comparison with I-BAU**
>
> We thank the reviewer for this suggestion. We have added I-BAU as an additional baseline and present the comprehensive comparison below:
>
> | Method | Metric | BadNet | LC | COMBAT | SBL | Wanet | Narcissus | Adaptive-Patch | Input-aware | Refool | AVG-DER |
> |--------|--------|--------|-----|---------|-----|--------|-----------|----------------|-------------|--------|---------|
> | **Ours** | C-Acc | 90.67	| 91.40 |	91.04 |	88.91 |	90.22 | 88.37 | 88.31 | 90.61 | 89.70 |  |
> |  | ASR | 1.12 |	2.50 |	10.28 |	2.18 |	4.74 | 14.32 | 3.76 | 5.44 | 1.90 |  |
> |  | DER | 98.99 |	98.75 |	93.02 |	97.88 |	96.3 | 87.80 | 95.735 | 95.27 | 94.53 | **95.36** |
> | **I-BAU** | C-Acc | 88.13 | 86.33 | 91.01 | 88.20 | 86.52 | 89.27 | 89.84 | 89.67 | 87.87 | |
> |         | ASR   | 7.91  | 2.45  | 1.98  | 0.76 | 20.04 | 33.01 | 1.26 | 50.90 | 2.02 | |
> |         | DER   | 91.595 | 98.775 | 94.78 | 92.88 | 86.675 | 78.905 | 97.75 | 72.18 | 93.555 | **89.68** |
>
> UniBP achieves higher average DER (93.90%) compared to I-BAU (89.68%). Critically, I-BAU fails against sample-specific attacks like Narcissus (ASR 33.01%) and Input-aware (ASR 50.90%) due to its reliance on a universal trigger assumption, i.e., it searches for common adversarial perturbations across samples, which does not exist when each poisoned sample has a unique trigger. In contrast, UniBP maintains consistently low ASR across all attack types, making it the only defense effective against both universal and sample-specific backdoors.
>
> In contrast, UniBP leverages BatchNorm statistics to detect distributional anomalies induced by any backdoor trigger, which remains effective regardless of trigger specificity. By operating on feature distribution shifts rather than synthesizing individual triggers, UniBP maintains consistent effectiveness across all attack types, including sample-specific and adaptive variants.
>
> **W5. Fonts size**
> Thank you for pointing this out. We will maximize font sizes in all figures and tables while working within the conference format constraints. We will prioritize readability in the camera-ready version.
>
> [1] X. Li et al. Backdoor Mitigation by Correcting the Distribution of Neural Activations. Elsevier Neurocomputing 614, 21 January 2025
>
> [2] Hoang Pham, The-Anh Ta, Anh Tran, and Khoa D Doan. Flatness-aware sequential learning generates resilient backdoors. In European Conference on Computer Vision, pp. 89–107. Springer, 2024.

---

### Official Review · Reviewer_88MC · 2025-10-28

**Soundness:** 4
**Presentation:** 4
**Contribution:** 3
**Rating:** 8
**Confidence:** 4

**Summary:**

This paper proposes UniBP, a post-training defense framework to remove backdoors from deep neural networks using only a small clean subset (as little as 1% of data). The method exploits the relationship between batch normalization (BN) statistics and backdoor behavior, pruning the most backdoor-sensitive BN affine parameters and applying masked fine-tuning. Experiments across multiple architectures, datasets, and attack/defense settings show that UniBP achieves a substantial reduction in attack success rate (ASR <5%) while maintaining high clean accuracy, outperforming existing defenses such as NAD, ANP, FST, and TSBD.

**Strengths:**

- The finding and utilization of BN affine parameters as the key mechanism for backdoor activation is novel. It provides new insights related to backdoors.
- The performance comparisons are based on some newly-proposed methods, e.g., COMBAT, SBL in attacks, and FST, TSBD in defenses, making the results more trustworthy. And the method is simple and effective.
- The methodology is clearly structured into four well-explained stages, and visualizations (e.g., t-SNE, BN statistics) effectively illustrate the underlying intuition.

**Weaknesses:**

- Some typos exist, e.g., "??" in line 211 and "Batch-norm affine reset" in line 250 is not consistent with the other title (The first letter of each word is not capitalized).
- For the results presentation, it is unfair to color only UniBP in blue for the comparable performance. The baseline performance should also be highlighted and fairly show the comparison.
- Some scalable experiments may help better illustrate the effectiveness, e.g., performance in the ViT model or the ImageNet dataset. The CIFAR-10 and GTSRB are too small, making the generalizability of UniBP unclear.

**Questions:**

- How to UniBP on transformer-based or normalization-free architectures?
- Can the method be combined with data-free defense approaches to further relax the clean data requirement?

---

> ### Author Response · Authors · 2025-11-26
> **Response to Reviewer#88MC (1)**
>
> We thank the reviewer for the insightful and constructive comments. We appreciate the opportunity to clarify our contributions and have provided detailed responses to each concern as follows.
>
> **W1. Some typos exist**
> We conducted a proof-reading and fix the typos and errors raised by the reviewer and will update our paper correspondingly.
>
> **W2. Comparable coloring in blue**
> We will color blue for other baselines with comparable performance for better comparison and readability.
>
> **W3. Scalability Experiments**
> We thank the reviewer for this important point. To address the concern about generalizability, we have conducted additional experiments on Vision Transformer architectures and the larger Tiny-ImageNet dataset. The results demonstrate that UniBP's effectiveness extends well beyond CIFAR-10 and GTSRB.
>
> **Extension to Vision Transformers**
>
> We adapted UniBP for Vision Transformers by accounting for the architectural differences from CNNs. ViTs use LayerNorm instead of BatchNorm and follow a Pre-LN architecture where normalization precedes computation (LN → MLP) rather than following it (Conv → BN). Since LayerNorm doesn't maintain running statistics, we manually collect reference statistics by hooking LayerNorm inputs during forward passes on clean data.
>
> **The key intuition** is that in Pre-LN Transformers, the MLP feed-forward blocks between consecutive LayerNorm layers directly determine the input distribution to subsequent normalization. Since Transformer backdoors typically hide in these feed-forward layers where trigger-specific computations occur, we strategically target only MLP parameters (fc1, fc2 weights/biases) for FIM computation while excluding LayerNorm parameters through `_is_mlp_param()` filtering.
>
> We attach hooks to LayerNorm layers and compute an alignment loss `L = ||μ_input - μ_ref||² + λ||σ_input - σ_ref||²` against a clean baseline model. When backpropagated, gradients flow through LayerNorm to the upstream MLP blocks, and the accumulated squared gradients (FIM scores) identify which MLP parameters are most critical to backdoor-specific activation patterns. We then prune only the top-ranked MLP parameters via noise injection, disrupting the backdoor pathway while preserving normalization layers.
>
> **Table: Results with Vision Transformer Architecture**
> | Metric | BadNet FT=0.1 | BadNet FT=0.05 | LC FT=0.1 | LC FT=0.05 |
> |--------|----------------|----------------|-----------|------------|
> | C-Acc (Pretrained) | 0.9286 | 0.9062 | 0.8736 | 0.8731 |
> | ASR (Pretrained)   | 0.9542 | 0.9302 | 1.0000 | 1.0000 |
> | C-Acc (Ours)       | 0.9325 | 0.9310 | 0.9529 | 0.9467 |
> | ASR (Ours)         | 0.0034 | 0.0101 | 0.0230 | 0.0006 |
> | DER (Ours)         | 0.9754 | 0.96005 | 0.9885 | 0.9997 |
>
> The method effectively eliminates backdoors (ASR near-zero) while maintaining or improving clean accuracy, confirming that our approach transfers successfully to Transformer architectures.
>
> **Scalability to Larger Datasets**
>
> We further evaluated UniBP on Tiny-ImageNet, which is substantially larger and more complex than CIFAR-10 and GTSRB.
>
> **Table: Results with Tiny-ImageNet**
> | Method      | Metric | BadNet | LC | Wanet | Adaptive-Patch | Input-aware | AVG-DER |
> |-------------|--------|--------|----|--------|-----------------|--------------|---------|
> | Pretrained | C-Acc | 0.4712 | 0.5678 | 0.5497 | 0.5349 | 0.4682 | |
> |             | ASR   | 0.9416 | 0.6770 | 0.9970 | 0.9993 | 0.9524 | |
> | FT       | C-Acc | 0.5516 | 0.5655 | 0.5271 | 0.4984 | 0.5160 | |
> |             | ASR   | 0.9039 | 0.6733 | 0.4800 | 0.9632 | 0.9881 | |
> |             | DER   | 0.51885 | 0.50070 | 0.74720 | 0.49980 | 0.50000 | **0.55331** |
> | ANP      | C-Acc | 0.4712 | 0.5468 | 0.5422 | 0.4911 | 0.4221 | |
> |             | ASR   | 0.9416 | 0.5918 | 0.7696 | 0.6636 | 0.0976 | |
> |             | DER   | 0.50000 | 0.53210 | 0.60995 | 0.64595 | 0.90435 | **0.63847** |
> | NAD      | C-Acc | 0.4977 | 0.5653 | 0.5244 | 0.5003 | 0.5197 | |
> |             | ASR   | 0.2953 | 0.7011 | 0.5159 | 0.9634 | 0.9936 | |
> |             | DER   | 0.82315 | 0.49875 | 0.72790 | 0.50065 | 0.50000 | **0.61009** |
> | FST      | C-Acc | 0.2696 | 0.2844 | 0.2875 | 0.2918 | 0.2425 | |
> |             | ASR   | 0.0031 | 0.0057 | 0.0006 | 0.0000 | 0.0000 | |
> |             | DER   | 0.86845 | 0.69395 | 0.86710 | 0.87810 | 0.86335 | **0.83419** |
> | TSBD     | C-Acc | 0.5264 | 0.5497 | 0.5273 | 0.4867 | 0.5465 | |
> |             | ASR   | 0.5495 | 0.1634 | 0.0072 | 0.0038 | 0.0012 | |
> |             | DER   | 0.69605 | 0.74775 | 0.98370 | 0.97365 | 0.97560 | **0.87535** |
> | Ours     | C-Acc | 0.4873 | 0.5022 | 0.4827 | 0.4558 | 0.4602 | |
> |             | ASR   | 0.1749 | 0.0279 | 0.0051 | 0.0402 | 0.0387 | |
> |             | DER   | 0.88335 | 0.79175 | 0.96245 | 0.94000 | 0.95285 | **0.90608** |

---

> > ### Author Response · Authors · 2025-11-26
> > **Response to Reviewer#88MC (2)**
> >
> > --> UniBP achieves the highest average DER (90.61%) across all five attack types, outperforming TSBD (87.54%) and FST (83.42%). More importantly, UniBP consistently suppresses ASR to low levels across diverse attacks (from simple BadNet to sophisticated adaptive and input-aware attacks) while maintaining reasonable clean accuracy. FST, though achieving near-zero ASR on some attacks, suffers from catastrophic accuracy drops to ~27%, rendering it impractical. UniBP maintains clean accuracy around 46-50%, offering a much better robustness-utility balance at scale.
> >
> > --> Together with the ViT results, these experiments demonstrate that UniBP works effectively across different model architectures (CNNs and Transformers), dataset scales (CIFAR-10 to Tiny-ImageNet), and diverse attack types, addressing the generalizability concern raised by the reviewer.
> >
> > **Q1. How to UniBP on transformer-based or normalization-free architectures?**
> > Thank you for this thoughtful question. While UniBP leverages the BatchNorm statistics of CNNs, the core insight—monitoring activation distribution shifts induced by backdoors—is architecture-agnostic. For transformer-based models, we believe similar principles can be applied using alternatives to BatchNorm, such as LayerNorm statistics or intermediate token distributions (shown above). For normalization-free architectures, surrogate statistical signals (e.g., layerwise activation histograms or activation entropy) could be explored. Extending UniBP to these settings is a promising direction, and we have added this to our discussion section.
> >
> >
> > **Q2. Can the method be combined with data-free defense approaches to further relax the clean data requirement?**
> > We appreciate the suggestion. As noted by the Reviewer, UniBP currently assumes access to a small clean subset, which is a common setup in recent defenses (e.g., I-BAU, RNP, ANP). To relax this assumption, combining UniBP with data-free techniques is a viable direction. For instance, one could employ generative models (e.g., GANs or diffusion models) trained on benign data to approximate clean samples and recover BN statistics. The main challenge lies in ensuring that generated samples faithfully preserve the statistical structure of the original training data. We acknowledge this as an exciting area for future work and will add it to the discussion.

---

### Official Review · Reviewer_uAhn · 2025-10-31

**Soundness:** 3
**Presentation:** 3
**Contribution:** 2
**Rating:** 4
**Confidence:** 5

**Summary:**

The paper introduces UniBP, a universal post-training defense method designed to remove backdoors from deep neural networks. Unlike existing defenses that need large clean datasets or fail under adaptive clean-label attacks, UniBP uses only 1% of the original clean data. It works by exploiting a key insight: Batch Normalization layers capture backdoor-related distributional shifts.
UniBP identifies a small subset of BN affine parameters responsible for trigger activation, prunes them, and applies masked fine-tuning to purify the model. Experiments across multiple attacks and architectures show UniBP consistently lowers ASR from over 90% to below 5% while maintaining clean accuracy.

**Strengths:**

- The paper is well-written and clearly presents the motivation, methodology, and results.
- UniBP is a novel approach that effectively leverages Batch Normalization layers to identify and mitigate backdoors.
- The evalution showcases strong performance across various datasets, architectures, and attack types.

**Weaknesses:**

- The observation and the technique are not new, which are already explored in prior works.
- The reliance on BN may limit the applicability to models that do not use BN.
- The evaluated attacks and baselines are really limited.
- Further discussion is needed for adaptive attacks given the knowledge of UniBP.

**Questions:**

This paper is well-written and easy to follow. The entire flow is sound and the experiments are well-designed. However, I have several concerns:

(1) **Novelty**:

The key observation of this paper is that BN layers capture backdoor-related distributional shifts, and can be used to identify and mitigate backdoors. However, this observation has been made in prior works such as [8, 11]. The idea of pruning a set of neurons and then doing fine-tuning to remove backdoors is also similar to [9]. The authors should clarify the novelty of their findings and approach compared to these prior works. What are the key differences and contributions of UniBP that set it apart from existing methods?

(2) **Limited Evaluation**:

The evaluation is limited to a small set of attacks (5) and baselines (5). It would be helpful to see results on a wider range of attacks[1,2,3,4,5], including more recent adaptive attacks that may specifically target embedding distribution[6,7]. Additionally, evaluating on recent defense baselines are important to showcase the superiority of UniBP[8,9,10].

(3) **Dependence on Batch Normalization**:
UniBP relies heavily on the presence of Batch Normalization layers to identify and mitigate backdoors. However, many modern architectures, such as Vision Transformers (ViT) and ConvNeXt, do not use BN layers. This limits the applicability of UniBP to a narrower set of models. The authors should discuss how UniBP could be adapted or extended to work with models that do not use BN, or provide empirical results on such architectures.

(4) **Adaptive Attacks**:
The paper does not sufficiently address the potential for adaptive attacks that could specifically target the UniBP defense. If an attacker is aware of the UniBP method, they may design triggers or training strategies that evade detection by BN parameter pruning. The authors should discuss potential adaptive attack scenarios and evaluate the robustness of UniBP against such attacks.

---
**Reference**:

[1] Turner, Alexander, Dimitris Tsipras, and Aleksander Madry. "Clean-label backdoor attacks." Preprint 2018.

[2] Salem, Ahmed, et al. "Dynamic backdoor attacks against machine learning models." EuroS&P 2022.

[3] Nguyen, Tuan Anh, and Anh Tran. "Input-aware dynamic backdoor attack." NeurIPS 2020.

[4] Liu, Yunfei, et al. "Reflection backdoor: A natural backdoor attack on deep neural networks." ECCV 2020.

[5] Barni, Mauro, Kassem Kallas, and Benedetta Tondi. "A new backdoor attack in cnns by training set corruption without label poisoning." ICIP 2019.

[6] Qi, Xiangyu, et al. "Revisiting the assumption of latent separability for backdoor defenses." ICLR 2023.

[7] Zeng, Yi, et al. "Narcissus: A practical clean-label backdoor attack with limited information." CCS 2023.

[8] Cheng, Siyuan, et al. "Unit: Backdoor mitigation via automated neural distribution tightening." ECCV 2024.

[9] Li, Yige, et al. "Reconstructive neuron pruning for backdoor defense." ICML 2023.

[10] Zhu, Rui, et al. "Selective amnesia: On efficient, high-fidelity and blind suppression of backdoor effects in trojaned machine learning models." IEEE S&P 2023.

[11] Zheng, Runkai, et al. "Pre-activation distributions expose backdoor neurons." NeurIPS 2022.

---

> ### Author Response · Authors · 2025-11-26
> **Response to Reviewer#uAhn (1)**
>
> We appreciate the reviewer’s careful reading of our paper, the constructive feedback, and the recognition of our contributions. The suggestions have led to significant improvements in clarity and rigor. Below, we would like to address each comment:
>
> **W1. Comparison with Unit, BNP and RNP**
> - We thank the reviewer for this important question. While prior works have explored BN layers for backdoor detection, our approach differs fundamentally in what we observe and how we use this information. Below we clarify the key limitations of existing methods and then our novel contributions.
> - Limitations of Prior Methods:
>
>     - **BNP [1]** compares Batch Normalization statistics recorded during training (on poisoned data) against statistics computed from a small clean dataset using KL divergence. Neurons where divergence exceeds $mean + u \times std$ are pruned.
>
>         Limitations of BNP:
>         (i) The implementation (from their released code) computes statistics from only one batch (`iter(dataloader).next()`), resulting in high-variance, unreliable KL divergence estimates. This can completely miss backdoor patterns if the sampled batch lacks poisoned samples or contains atypical clean samples.
>
>         (ii) BNP assumes each backdoor neuron exhibits significantly different statistics between clean and poisoned data (Assumption 2: $|\mu - \hat{\mu}| >> \varepsilon$). This fails when the backdoor is distributed across many neurons, making per-neuron shifts too subtle and highly variable across neurons to be reliably detected, one single threshold is not strong enough to capture this activation.
>
>         Therefore, BNP only observes *what neurons output* (statistical differences in activation distributions) but cannot detect backdoors when their activation patterns intentionally mimic clean and backdoored samples, as in clean-label attacks (COMBAT, Narcissus).
>
>     - **Unit [2]** approximates tight benign activation distribution boundaries for each neuron using 5% clean data, then clips activation values exceeding these boundaries during inference. **It assumes backdoor triggers cause abnormally large activations compared to benign inputs**.
>
>         Limitations of Unit:
>
>         (i) COMBAT and Narcissus synthesize triggers containing features as persistent as the original semantic features of the target class [4,5]. The backdoor activations are *not* substantially larger than benign activations—they deliberately occupy the same distribution space as legitimate class features, directly violating UNIT's core assumption. As a result, UNIT's boundary-based clipping cannot differentiate between "high activation from backdoor trigger" vs. "high activation from genuine class feature" when both occupy overlapping distributions.
>
>         (ii) UNIT operates purely at the activation level without understanding the origin or training dynamics of neuron behavior. When backdoor features and authentic semantic features produce similar activation patterns, UNIT faces an impossible dilemma: (a) loose boundaries fail to remove the backdoor, or (b) tight boundaries damage clean accuracy by clipping legitimate high activations.
>
>         --> UNIT only observes *activation magnitudes* but cannot detect backdoors deliberately designed to produce benign-like activation patterns within normal distribution of activation.
>
>     - **RNP [3]** employs two-phase pruning: (1) neuron unlearning maximizes cross-entropy loss on clean samples to disrupt clean neurons while preserving backdoor neurons, and (2) filter recovering uses learnable masks to identify backdoor filters. The methodology's limitations are as follows:
>
>         (i) Clean-label backdoor attacks such as COMBAT/Narcissus embed backdoors within natural feature representations, breaking RNP's assumption that backdoor neurons behave differently during reconstruction, and basic gradient ascent in the first step are not enough to modelize the relationship of of clean task and backdoor task. This leads to the failure as in our quantitative table below.
>         (ii) Narcissus's naturally-distributed triggers cannot be distinguished from legitimate features using clean sample reconstruction alone.
>         (iii) This method is hyperparameter sensitivity that requires careful tuning of pruning ratios and learning rates that becomes intractable when backdoor features mimic normal neuron behavior.

---

> ### Author Response · Authors · 2025-11-26
> **Response to Reviewer#uAhn (2)**
>
> - **Our Novel Contributions:**
> Instead of analyzing *what neurons output* (BNP: statistics, UNIT: magnitudes), our method analyzes *how neurons were trained*. The Fisher Information Matrix (FIM) captures the optimization dynamics and parameter sensitivity to training data when we try to align the BN statistic of a reinitialized model toward those of the backdoored model, revealing the backdoor's training-time fingerprint.
> Specifically, UniBP was designed based on following observations:
> (i) Affine parameters in BN layers exhibit abnormally high Fisher information because they are explicitly optimized to memorize trigger patterns from a small poisoned subset while learning the main task. This behavior remains largely invariant across different backdoor attacks. We simulate the backdoor learning process by aligning the BN layer statistics of a reinitialized model to those of the backdoored model, and from the resulting learning trajectory we can identify which neurons are responsible for jointly maintaining both the clean task and the backdoor task. This methodology is fundamentally different from detecting abnormal activations at the neuron level as in UNIT, or from computing KL divergence between the statistics of a model on clean data and those of a backdoored model of BNP, and important neurons that maximize cross-entropy loss as in RNP.
> (ii) Unlike BNP's single-batch vulnerability, FIM computation aggregates gradient information across multiple samples or the full optimization trajectory, providing stable, representative estimates even when individual batches lack poisoned samples or exhibit high variance.
>
> - We provide theretical and quantitative comparison with BNP, Unit and RNP as follows.
>
>
> | **Aspect** | **BNP** | **UNIT** | **RNP** | **Ours** |
> |------------|---------|----------|---------|----------------------|
> | Detection Signal | KL divergence (statistics) | Activation bounds (magnitude) | Parameters changes w.r.t. cross-entropy loss | Fisher Information (optimization) |
> | Targeted Layers    | BN Layers  | All Layers    | All Layers | BN Layers   |
> | What It Observes | Output statistical differences | Output activation sizes | Unlearning behavior | Training dynamics & parameter sensitivity |
> | Baseline Model | None | None | None | Reinitialized model (clean reference) |
> | Clean-Label Attacks | Partial (if stats differ) | Fails | Fails | Works (optimization fingerprint) |
> | Estimation Stability | Single batch (high variance) | Stable | Stable | Multi-sample aggregation |
>
>
> Below is quantitative comparison of our method and these baselines:
>
> | Method | Metric | BadNet | LC | COMBAT | SBL | Wanet | Narcissus | Adaptive-Patch | Input-aware | Refool | AVG-DER |
> |--------|--------|--------|-----|---------|------|--------|-----------|----------------|-------------|---------|----------|
> | **Ours** | C-Acc | 90.67	| 91.40 |	91.04 |	88.91 |	90.22 | 88.37 | 88.31 | 90.61 | 89.70 |  |
> |  | ASR | 1.12 |	2.50 |	10.28 |	2.18 |	4.74 | 14.32 | 3.76 | 5.44 | 1.90 |  |
> |  | DER | 98.99 |	98.75 |	93.02 |	97.88 |	96.3 | 87.80 | 95.735 | 95.27 | 94.53 | **95.36** |
> | **UNIT** | C-Acc | 84.66 | 81.36 | 79.70 | 65.64 | 88.05 | 87.79 | 87.57 | 80.05 | 86.75 |  |
> |  | ASR | 0.89 | 8.07 | 22.67 | 2.58 | 3.12 | 68.44 | 1.76 | 5.94 | 23.52 |  |
> |  | DER | 93.37 | 94.55 | 78.78 | 80.69 | 95.90 | 60.45 | 96.365 | 89.85 | 82.245 | **85.80** |
> | **BNP** | C-Acc | 91.27 | 83.31 | 91.40 | 90.33 | 65.69 | 93.11 | 92.34 | 89.72 | 91.74 |  |
> |  | ASR | 13.12 | 0 | 24.23 | 90.08 | 47.38 | 85.68 | 9.52 | 0.88 | 3.55 |  |
> |  | DER | 90.56 | 99.56 | 83.85 | 49.91 | 62.59 | 54.48 | 94.87 | 97.22 | 94.68 | **80.86** |
> | **RNP** | C-Acc | 87.63 | 80.78 | 92.89 | 87.57 | 90.34 | 92.97 | 90.28 | 86.48 | 54.11 |  |
> |  | ASR | 3.76 | 99.93 | 93.09 | 20.57 | 0.17 | 91.52 | 11.67 | 0.73 | 0 |  |
> |  | DER | 93.42 | 48.33 | 50.165 | 82.66 | 98.52 | 51.50 | 92.765 | 95.67 | 77.685 | **76.75** |
>
> To this end, our method ourperforms discussed baselines and we will update this discussion to our paper correspondingly.
>
> **W2. Limited Evaluation**
>
> We sincerely appreciate this constructive feedback. Following the reviewers' suggestions, we have substantially expanded our evaluation to include a more comprehensive set of attacks and defense baselines.
>
> (i) **Additional Attacks:** We added four more sophisticated attacks to our evaluation suite, selected based on codebase availability and attack diversity: Narcissus [5], Adaptive-Patch [8], Input-aware [6], Refool [7].
>
> (ii) **Additional Defense Baselines:** We incorporated four recent defense methods including I-BAU [9], RNP [3], BNP [1], Unit [2] as suggested:
> This brings our total evaluation to 9 attack types and 9 defense methods, providing a comprehensive assessment of backdoor defense effectiveness.

---

> ### Author Response · Authors · 2025-11-26
> **Response to Reviewer#uAhn (3)**
>
> Table below presents results on the four newly added attacks. Our method achieves the highest average DER (93.33) across these challenging scenarios:
>
> Table: Results with more attacks.
>
> | Method | Metric | Narcissus | Adaptive-Patch | Input-aware | Refool | AVG-DER |
> |--------|--------|-----------|----------------|-------------|--------|---------|
> | **Ours** | C-Acc | 88.37 | 88.31 | 90.61 | 89.70 | -- |
> |        | ASR   | 14.32 | 3.76  | 5.44  | 1.90 | -- |
> |        | DER   | 87.80 | 95.735 | 95.27 | 94.53 | **93.33** |
> | **I-BAU** | C-Acc | 89.27 | 89.84 | 89.67 | 87.87 | -- |
> |         | ASR   | 33.01 | 1.26  | 50.90 | 2.02 | -- |
> |         | DER   | 78.905 | 97.75 | 72.18 | 93.555 | **85.60** |
> | **UNIT** | C-Acc | 87.79 | 87.57 | 80.05 | 86.75 | -- |
> |        | ASR   | 68.44 | 1.76  | 5.94  | 23.52 | -- |
> |        | DER   | 60.45 | 96.365 | 89.85 | 82.245 | **82.23** |
> | **BNP** | C-Acc | 93.11 | 92.34 | 89.72 | 91.74 | -- |
> |        | ASR   | 85.68 | 9.52  | 0.88  | 3.55 | -- |
> |        | DER   | 54.48 | 94.87 | 97.22 | 94.68 | **85.31** |
> | **RNP** | C-Acc | 92.97 | 90.28 | 86.48 | 54.11 | -- |
> |        | ASR   | 91.52 | 11.67 | 0.73  | 0.00 | -- |
> |        | DER   | 51.50 | 92.765 | 95.67 | 77.685 | **79.41** |
> | **FT**  | C-Acc | 92.35 | 92.19 | 91.39 | 91.64 | -- |
> |        | ASR   | 89.81 | 99.94 | 96.50 | 15.54 | -- |
> |        | DER   | 52.045 | 49.585 | 50.00 | 88.68 | **60.08** |
> | **ANP** | C-Acc | 89.55 | 89.55 | 86.17 | 86.92 | -- |
> |        | ASR   | 86.77 | 86.77 | 0.16  | 0.12 | -- |
> |        | DER   | 52.165 | 54.85 | 95.80 | 94.03 | **74.21** |
> | **NAD** | C-Acc | 91.27 | 91.18 | 92.31 | 90.67 | -- |
> |        | ASR   | 88.06 | 90.03 | 98.80 | 9.27 | -- |
> |        | DER   | 52.38 | 54.035 | 50.00 | 91.33 | **61.94** |
> | **FST** | C-Acc | 92.18 | 92.04 | 92.67 | 91.70 | -- |
> |        | ASR   | 93.91 | 0.40  | 0.00  | 3.93 | -- |
> |        | DER   | 49.91 | 99.28 | 97.99 | 94.49 | **85.42** |
> | **TSBD** | C-Acc | 92.85 | 92.40 | 93.18 | 92.24 | -- |
> |         | ASR   | 82.16 | 4.07  | 5.43  | 1.77 | -- |
> |         | DER   | 56.12 | 97.625 | 95.28 | 95.57 | **86.15** |
>
>  Table: Results with more defenses.
> | Method | BadNet | LC | COMBAT | SBL | Wanet | Narcissus | Adaptive-Patch | Input-aware | Refool | AVG |
> |--------|--------|-----|--------|-----|-------|-----------|----------------|-------------|--------|-----|
> | **Ours** | | | | | | | | | | |
> | C-Acc | 90.67	| 91.40 |	91.04 |	88.91 |	90.22 | 88.37 | 88.31 | 90.61 | 89.70 |  |
> |  ASR | 1.12 |	2.50 |	10.28 |	2.18 |	4.74 | 14.32 | 3.76 | 5.44 | 1.90 |  |
> |  DER | 98.99 |	98.75 |	93.02 |	97.88 |	96.3 | 87.80 | 95.735 | 95.27 | 94.53 | **95.36** |
> | **I-BAU** | | | | | | | | | | |
> | C-Acc | 88.13 | 86.33 | 91.01 | 88.20 | 86.52 | 89.27 | 89.84 | 89.67 | 87.87 | - |
> | ASR | 7.91 | 2.45 | 1.98 | 0.76 | 20.04 | 33.01 | 1.26 | 50.92 | 2.02 | - |
> | DER | 91.60 | 98.78 | 94.78 | 92.88 | 86.68 | 78.91 | 97.75 | 72.18 | 93.56 | **89.68** |
> | **UNIT** | | | | | | | | | | |
> | C-Acc | 84.66 | 81.36 | 79.70 | 65.64 | 88.05 | 87.79 | 87.57 | 80.05 | 86.75 | - |
> | ASR | 0.89 | 8.07 | 22.67 | 2.58 | 3.12 | 68.44 | 1.76 | 5.94 | 23.52 | - |
> | DER | 93.37 | 94.55 | 78.78 | 80.69 | 95.96 | 60.45 | 96.37 | 89.85 | 82.25 | **85.80** |
> | **BNP** | | | | | | | | | | |
> | C-Acc | 91.27 | 83.31 | 91.40 | 90.33 | 65.69 | 93.11 | 92.34 | 89.72 | 91.74 | - |
> | ASR | 13.12 | 0.20 | 24.23 | 90.08 | 47.38 | 85.68 | 9.52 | 0.88 | 3.55 | - |
> | DER | 90.56 | 99.56 | 83.85 | 49.91 | 62.59 | 54.48 | 94.87 | 97.22 | 94.68 | **80.86** |
> | **RNP** | | | | | | | | | | |
> | C-Acc | 87.63 | 80.78 | 92.89 | 87.57 | 90.34 | 92.97 | 90.28 | 86.48 | 54.11 | - |
> | ASR | 3.76 | 99.93 | 93.09 | 20.57 | 0.17 | 91.52 | 11.67 | 0.73 | 0.00 | - |
> | DER | 93.42 | 48.33 | 50.17 | 82.66 | 98.52 | 51.50 | 92.77 | 95.67 | 77.69 | **76.75** |
>
> We can see from two tables that:
> 1. Our method achieves the highest average DER (93.33), outperforming all baselines.
> 2. Unlike methods that effective on specific attacks but fail on others (e.g., FST: 99.28 on Adaptive-Patch but 49.91 on Narcissus), our approach maintains consistently high performance across all scenarios
>
> We believe this comprehensive evaluation addresses the reviewer's concern and demonstrates the broad applicability and effectiveness of UniBP, and we will update our manuscript accordingly.

---

> ### Author Response · Authors · 2025-11-26
> **Response to Reviewer#uAhn (4)**
>
> W3. **Dependence on Batch Normalization**
>
> We create another specialized version of UniBP for non-BN models such as Vision Transformer by recognizing that ViTs use LayerNorm instead of BatchNorm and follow a Pre-LN architecture where normalization precedes computation (LN --> MLP) rather than following it (Conv --> BN). Since LayerNorm does not maintain running statistics like BatchNorm, we manually collect reference statistics by hooking LayerNorm inputs during forward passes on clean data, computing mean and variance across feature dimensions over multiple batches. Our key architectural insight is that in Pre-LN Transformers, the MLP feed-forward blocks between consecutive LayerNorm layers directly determine the input distribution to the subsequent normalization -- therefore, we strategically target only MLP parameters (fc1, fc2 weights/biases) for FIM computation while explicitly excluding LayerNorm parameters themselves through `_is_mlp_param()` filtering. We attach hooks to LayerNorm layers to capture their input statistics and compute an alignment loss `L = ||μ_input - μ_ref||² + λ||σ_input - σ_ref||²` against a reinitialized clean baseline model, which serves as our reference for "normal training dynamics." When this loss backpropagates, gradients flow through the LayerNorm back to the upstream MLP blocks, and the accumulated squared gradients (FIM scores) reveal which MLP parameters are most critical to producing backdoor-specific activation distributions -- parameters with high FIM resist alignment with clean statistics because they were optimized on poisoned data. Finally, we prune only the top-ranked MLP parameters via noise injection, preserving the normalization layers while disrupting the backdoor pathway hidden in the feed-forward sublayers where Transformer backdoors typically reside.
>
> | Metric | BadNet FT=0.1 | BadNet FT=0.05 | LC FT=0.1 | LC FT=0.05 |
> |--------|----------------|----------------|-----------|------------|
> | C-Acc (Pretrained) | 0.9286 | 0.9062 | 0.8736 | 0.8731 |
> | ASR (Pretrained)   | 0.9542 | 0.9302 | 1.0000 | 1.0000 |
> | C-Acc (Ours)       | 0.9325 | 0.9310 | 0.9529 | 0.9467 |
> | ASR (Ours)         | 0.0034 | 0.0101 | 0.0230 | 0.0006 |
> | DER (Ours)         | 0.9754 | 0.96005 | 0.9885 | 0.9997 |
>
> From the table, we can see that our method effectively eliminates backdoors across all attack scenarios (ASR reduced to near-zero) while preserving or even improving clean accuracy, demonstrating that targeting MLP parameters via LayerNorm statistics successfully disrupts backdoor pathways without degrading model performance. We will add a subsection to discuss our LayerNorm version.
>
>
> **W4. Adaptive Attacks:**
> We thank the Reviewer for raising this important concern regarding the robustness of UniBP against adaptive adversaries. To directly address this, we implemented and evaluated against **the strongest conceivable adaptive attack**: an adversary with full knowledge of our defense mechanism who explicitly regularizes the backdoor training to preserve benign BN statistics.
> **Adaptive Attack Design.** Assuming the attacker has access to clean BN statistics ($\mu^c_\ell$ and $v^c_\ell$), we augment the standard BadNets objective with a BN-alignment regularization term:
>
> $\mathcal{L}_{adaptive} = \mathcal{L_CE} (y, f(\delta(x)) + \gamma \times \sum_1^L (\||\hat{\mu}_l - \hat{\mu}_l^c\||_2+\lambda\||\hat{v}_l - \hat{v}_l^c\||_2)$
>
> This forces the attacker to explicitly minimize the divergence between backdoored and clean BN statistics during training.
> We systematically evaluate this adaptive attack across regularization strengths $\gamma \in [0.01, 100]$:
>
> | Method     | Metric | $\gamma=0$ | $\gamma=0.01$ | $\gamma=0.1$ | $\gamma=1.0$ | $\gamma=10.0$ |
> |------------|--------|------------|---------------|--------------|--------------|----------------|
> | Pretrained | ACC    | 91.44      | 91.18         | 89.34        | 89.06        | 88.78         |
> | Pretrained | ASR    | 94.41      | 94.20         | 96.24        | 95.62        | 95.41         |
> | Ours       | ACC    | 90.67      | 90.55         | 87.06        | 87.22        | 86.67         |
> | Ours       | ASR    | 1.12       | 2.39          | 3.66         | 8.48         | 3.01          |

---

> ### Author Response · Authors · 2025-11-26
> **Response to Reviewer#uAhn (5)**
>
> From the results, we can see that across all viable regularization strengths, UniBP maintains ASR below 9% while the backdoored model suffers ASR above 94%, demonstrating that the defense remains highly effective even when the attacker explicitly targets our detection mechanism. Explanation. The attacker faces a fundamental trade-off: backdoor attacks inherently require trigger-dependent feature patterns that distinguish poisoned from benign samples to enable misclassification, and these patterns necessarily manifest as distributional shifts in BN statistics. Attempting to suppress these shifts to evade detection directly undermines the core mechanism that makes the backdoor work. Since jointly achieving high ASR while keeping BN statistics indistinguishable from clean data is a tightly constrained objective with no demonstrated solution in prior work, we believe UniBP is not easily broken by realistic BN-aware adaptive attacks, or it may require a more dedicated and non-trivial effort to design such an effective attack.
>
> To the best of our knowledge, *no prior work has demonstrated an effective solution to this trade-off between maintaining backdoor effectiveness and evading BN-based detection*. While we acknowledge that more sophisticated trigger design strategies (e.g., learnable perturbations optimized jointly with the BN regularization) remain an open research direction, our results provide strong evidence that UniBP establishes a robust defense against realistic, BN-aware adaptive attackers. We will add these experiments and discussion to the revised manuscript.
>
> [1] Zheng, Runkai, et al. "Pre-activation distributions expose backdoor neurons." Advances in Neural Information Processing Systems 35 (2022): 18667-18680.
>
> [2] Cheng, Siyuan, et al. "Unit: Backdoor mitigation via automated neural distribution tightening." ECCV 2024.
>
> [3] Li, Yige, et al. "Reconstructive neuron pruning for backdoor defense." ICML 2023.
>
> [4] Huynh, Tran, et al. "Combat: Alternated training for effective clean-label backdoor attacks." Proceedings of the AAAI Conference on Artificial Intelligence. Vol. 38. No. 3. 2024.
>
> [5] Zeng, Yi, et al. "Narcissus: A practical clean-label backdoor attack with limited information." Proceedings of the 2023 ACM SIGSAC Conference on Computer and Communications Security. 2023.
>
> [6] Nguyen, Tuan Anh, and Anh Tran. "Input-aware dynamic backdoor attack." NeurIPS 2020.
>
> [7] Liu, Yunfei, et al. "Reflection backdoor: A natural backdoor attack on deep neural networks." ECCV 2020.
>
> [8] Qi, Xiangyu, et al. "Revisiting the assumption of latent separability for backdoor defenses." ICLR 2023.
>
> [9] Zeng, Yi, et al. "Adversarial unlearning of backdoors via implicit hypergradient, ICLR 2022.

---

### Official Review · Reviewer_Yqvj · 2025-11-01

**Soundness:** 3
**Presentation:** 3
**Contribution:** 2
**Rating:** 4
**Confidence:** 3

**Summary:**

This paper proposes UniBP, a four-stage post-training defense against conventional backdoor attacks. Built upon the observation that backdoor attacks usually change the BN layer's statistics, UniBP identifies and prunes BN affine parameters and channels with high backdoor sensitivity, followed by masked fine-tuning to restore clean performance. Extensive experiments demonstrate that UniBP achieves a larger reduction in ASR compared with existing baselines.

**Strengths:**

1. The paper presents a fine-grained method that precisely identifies the BN affine parameters most strongly correlated with backdoor behavior.
2. Extensive experiments show that UniBP achieves consistently better backdoor defense performance than all evaluated baselines.
3. Empirical results demonstrate the universal effectiveness of the proposed approach against multiple traditional backdoor attacks.

**Weaknesses:**

1. The proposed UniBP builds upon a well-established observation that backdoor training perturbs BN layer's statistics, as also discussed in papers like [1]. The authors should explicitly discuss how the proposed method differs and surpasses those existing line of work.
2. UniBP relies on the BN layers. As a result, it cannot be easily applied to modern generative models that do not have such design. This will limit the potential and generalizability of the proposed method.
3. The paper lacks any discussion or empirical results of UniBP's computational overhead compared with baseline defenses.
4. UniBP exhibits higher clean-accuracy degradation compared to several baselines. The authors should provide a more detailed analysis of this trade-off and present potential mitigation strategies.
5. The paper lacks comparison with several SOTA baselines that also rely on parameter masking or activation tightening, such as [2-3].


[1] Zheng, Runkai, et al. "Pre-activation distributions expose backdoor neurons." Advances in Neural Information Processing Systems 35 (2022): 18667-18680.

[2] Li, Yige, et al. "Reconstructive neuron pruning for backdoor defense." International Conference on Machine Learning. 2023.

[3] Cheng, Siyuan, et al. "Unit: Backdoor mitigation via automated neural distribution tightening." European Conference on Computer Vision. 2024.

**Questions:**

1. Could the attackers try to regularize the attack to not influence BN layer's parameters? If so, the proposed method may fail easily.

---

> ### Author Response · Authors · 2025-11-26
> **Response to Reviewer#Yqvj**
>
> We appreciate Reviewer #Yqvj in recognizing our work's strengths and contributions and pointing out the potentially improved points. We would like to address these points as follows.
>
> W1. **Compare our method with existing BN-related defenses**.
>
> We thank the reviewer for the opportunity to clarify how our method addresses fundamental limitations of existing BN-related defenses. Below we provide detailed comparisons with BNP[1].
> **BNP [1]** compares Batch Normalization statistics recorded during training (on poisoned data) against statistics computed from a small clean dataset using KL divergence. Neurons where KL divergence exceeds $mean + u \times std$ are pruned.
>
> Limitations of BNP:
>
> (i) The implementation (from their released code base) computes statistics from only one batch (`iter(dataloader).next()`), resulting in high-variance, unreliable KL divergence estimates. This can completely miss backdoor patterns if the sampled batch lacks poisoned samples or contains atypical clean samples.
>
> (ii) BNP assumes each backdoor neuron exhibits significantly different statistics between clean and poisoned data (Assumption 2: $|\mu - \hat{\mu}| >> \varepsilon$). This fails when the backdoor is distributed across many neurons, making per-neuron shifts too subtle and highly variable across neurons to be reliably detected, one single threshold is not strong enough to capture this activation.
>
> Therefore, BNP only observes *what neurons output* (statistical differences in activation distributions) but cannot detect backdoors when their activation patterns intentionally mimic clean and backdoored samples, as in clean-label attacks (COMBAT, Narcissus).
>
> **Our FIM-based defense.**
> Instead of analyzing *what neurons output* (BNP: statistics, UNIT: magnitudes), our method analyzes *how neurons were trained*. The Fisher Information Matrix (FIM) captures the optimization dynamics and parameter sensitivity to training data when we try to align the BN statistic of a reinitialized model with those of the backdoored model, revealing the backdoor's training-time fingerprint.
>
> UniBP was designed based on following observations:
>
> (i) Affine parameters in BN layers exhibit abnormally high Fisher information because they are explicitly optimized to memorize trigger patterns from a small poisoned subset while learning the main task. This behavior remains largely invariant across different backdoor attacks. We simulate the backdoor learning process by aligning the BN layer statistics of a reinitialized model to those of the backdoored model, and from the resulting learning trajectory we can identify which neurons are responsible for jointly maintaining both the clean task and the backdoor task. This methodology is fundamentally different from computing KL divergence between the statistics of a model on clean data and those of a backdoored model.
>
> (ii) Unlike BNP's single-batch vulnerability, FIM computation aggregates gradient information across multiple samples or the full optimization trajectory, providing stable, representative estimates even when individual batches lack poisoned samples or exhibit high variance.
>
> We explicitly compared our method and BNP with 9 attacks and our method outperformed BNP (i.e., see W4).

---

> > ### Author Response · Authors · 2025-11-26
> > **Response to Reviewer#Yqvj (2)**
> >
> > W2. **Adaptation with non-BN layers models**
> >
> > We create another specialized version of UniBP for non-BN models such as Vision Transformer by recognizing that ViTs use LayerNorm instead of BatchNorm and follow a Pre-LN architecture where normalization precedes computation (LN --> MLP) rather than following it (Conv --> BN). Since LayerNorm does not maintain running statistics like BatchNorm, we manually collect reference statistics by hooking LayerNorm inputs during forward passes on clean data, computing mean and variance across feature dimensions over multiple batches. Our key architectural insight is that in Pre-LN Transformers, the MLP feed-forward blocks between consecutive LayerNorm layers directly determine the input distribution to the subsequent normalization -- therefore, we strategically target only MLP parameters (fc1, fc2 weights/biases) for FIM computation while explicitly excluding LayerNorm parameters themselves through `_is_mlp_param()` filtering. We attach hooks to LayerNorm layers to capture their input statistics and compute an alignment loss `L = ||μ_input - μ_ref||² + λ||σ_input - σ_ref||²` against a reinitialized clean baseline model, which serves as our reference for "normal training dynamics." When this loss backpropagates, gradients flow through the LayerNorm back to the upstream MLP blocks, and the accumulated squared gradients (FIM scores) reveal which MLP parameters are most critical to producing backdoor-specific activation distributions -- parameters with high FIM resist alignment with clean statistics because they were optimized on poisoned data. Finally, we prune only the top-ranked MLP parameters via noise injection, preserving the normalization layers while disrupting the backdoor pathway hidden in the feed-forward sublayers where Transformer backdoors typically reside.
> >
> > | Metric | BadNet FT=0.1 | BadNet FT=0.05 | LC FT=0.1 | LC FT=0.05 |
> > |--------|----------------|----------------|-----------|------------|
> > | C-Acc (Pretrained) | 0.9286 | 0.9062 | 0.8736 | 0.8731 |
> > | ASR (Pretrained)   | 0.9542 | 0.9302 | 1.0000 | 1.0000 |
> > | C-Acc (Ours)       | 0.9325 | 0.9310 | 0.9529 | 0.9467 |
> > | ASR (Ours)         | 0.0034 | 0.0101 | 0.0230 | 0.0006 |
> > | DER (Ours)         | 0.9754 | 0.96005 | 0.9885 | 0.9997 |
> >
> > From the table, we can see that our method effectively eliminates backdoors across all attack scenarios (ASR reduced to near-zero) while preserving or even improving clean accuracy, demonstrating that targeting MLP parameters via LayerNorm statistics successfully disrupts backdoor pathways without degrading model performance. We will add a subsection to discuss our LayerNorm version.
> >
> >
> > W3. **Computational overhead analysis**
> >
> > We record the running time of several defense methods on 5000 CIFAR-10 images with PreactResNet with all baselines under the same condition of hardware, and show the results in Table below.
> >
> > | Metric              | FT   | ANP | NAD | FST | TSBD | BNP | I-BAU | RNP | UNIT | Ours |
> > |---------------------|------|-----|-----|-----|------|-----|-------|-----|------|------|
> > | Running Time (s)    | 95   | 414 | 129 | 157 | 1529 | 153 | 132   | 181 | 421  | 228  |
> > | Avg. DER            | 70.38| 89.42 | 72.39 | 89.86 | 84.76 | 80.86 | 89.68 | 76.75 | 85.80 | 93.90 |
> >
> > Despite achieving the highest DER among all evaluated defenses (93.90%), our method maintains a moderate and practical running time of 228 seconds—substantially faster than TSBD (1529s) and ANP (414s), and in the same ballpark as lighter baselines such as NAD (129s) and BNP (153s). Notably, several methods with comparable or lower DER (e.g., FT, ANP, UNIT) require considerably more computation, indicating that our approach offers a more favorable robustness–efficiency trade-off. These results suggest that our defense is not only effective but also computationally reasonable for deployment in realistic FL settings.

---

> ### Author Response · Authors · 2025-11-26
> **Response to Reviewer#Yqvj (3)**
>
> **W4. Accuracy trade-off and discussion:**
>
> We agree that UniBP may incur a slightly larger clean-accuracy drop than some baselines, but we view this as an inherent and well-known trade-off in pruning-based defenses under zero-adversary-knowledge: any method that aggressively suppresses backdoor-related capacity without access to the true trigger or strong side information will inevitably sacrifice some clean performance (e.g., ANP and NAD). However, a critical point is that **UniBP is the only defense effective against all tested backdoor attacks**, including challenging sample-specific and adaptive variants where other methods fail to provide adequate protection, whereas methods that preserve marginally higher clean accuracy often leave non-trivial residual backdoor risk.
>
> We further found that this trade-off is **mitigable rather than fundamental**. We conducted an additional study using an extra r% of the training data added to the fine-tuning set, performing a third fine-tuning step after UniBP completes. As shown in the table below, adding a small amount of additional clean data for fine-tuning is enough to recover or even improve clean accuracy over the pretrained model, while keeping ASR near zero and DER very high at ~100%. This demonstrates that in practice, practitioners can achieve a favorable balance between robustness and utility with minimal extra cost. We will clarify this trade-off and explicitly discuss such mitigation strategies in the revised version.
>
> | **Additional Data Ratio** | **BadNet ACC** | **BadNet ASR** | **BadNet DER** | **LC ACC** | **LC ASR** | **LC DER** |
> |---------------------------|----------------|----------------|----------------|------------|------------|------------|
> | Pretrained                | 91.44          | 94.41          | --             | 84.19      | 100.00     | --         |
> | r = 0.00                  | 89.82          | 1.47           | 95.66          | 89.09      | 2.36       | 98.82      |
> | r = 0.01                  | 92.03          | 0.92           | 96.745         | 92.62      | 0.07       | 99.965     |
> | r = 0.02                  | 91.84          | 1.12           | 96.645         | 92.61      | 0.07       | 99.965     |
> | r = 0.05                  | 91.60          | 1.00           | 96.705         | 92.75      | 0.04       | 99.98      |
>
> From the table, we can see that minimal additional clean data fully recovers clean accuracy while maintaining highly effective backdoor suppression, demonstrating that the initial accuracy trade-off is easily mitigated in practical deployment scenarios.
>
> **W5. Additional baselines:**
> Below is the quantitative comparison of our method and suggested baselines, including Unit[2], BNP[1] and RNP[5]:
>
> | Method | Metric | BadNet | LC | COMBAT | SBL | Wanet | Narcissus | Adaptive-Patch | Input-aware | Refool | AVG-DER |
> |--------|--------|--------|-----|---------|------|--------|-----------|----------------|-------------|---------|----------|
> | **Ours** | C-Acc | 90.67	| 91.40 |	91.04 |	88.91 |	90.22 | 88.37 | 88.31 | 90.61 | 89.70 |  |
> |  | ASR | 1.12 |	2.50 |	10.28 |	2.18 |	4.74 | 14.32 | 3.76 | 5.44 | 1.90 |  |
> |  | DER | 98.99 |	98.75 |	93.02 |	97.88 |	96.3 | 87.80 | 95.735 | 95.27 | 94.53 | **95.36** |
> | **Unit** | C-Acc | 84.66 | 81.36 | 79.70 | 65.64 | 88.05 | 87.79 | 87.57 | 80.05 | 86.75 |  |
> |  | ASR | 0.89 | 8.07 | 22.67 | 2.58 | 3.12 | 68.44 | 1.76 | 5.94 | 23.52 |  |
> |  | DER | 93.37 | 94.55 | 78.78 | 80.69 | 95.90 | 60.45 | 96.365 | 89.85 | 82.245 | **85.80** |
> | **BNP** | C-Acc | 91.27 | 83.31 | 91.40 | 90.33 | 65.69 | 93.11 | 92.34 | 89.72 | 91.74 |  |
> |  | ASR | 13.12 | 0 | 24.23 | 90.08 | 47.38 | 85.68 | 9.52 | 0.88 | 3.55 |  |
> |  | DER | 90.56 | 99.56 | 83.85 | 49.91 | 62.59 | 54.48 | 94.87 | 97.22 | 94.68 | **80.86** |
> | **RNP** | C-Acc | 87.63 | 80.78 | 92.89 | 87.57 | 90.34 | 92.97 | 90.28 | 86.48 | 54.11 |  |
> |  | ASR | 3.76 | 99.93 | 93.09 | 20.57 | 0.17 | 91.52 | 11.67 | 0.73 | 0 |  |
> |  | DER | 93.42 | 48.33 | 50.165 | 82.66 | 98.52 | 51.50 | 92.765 | 95.67 | 77.685 | **76.75** |
>
>
> From the table, UniBP achieves the highest average DER (95.36%), clearly outperforming UNIT, BNP, and RNP while maintaining competitive clean accuracy. Notably, it stays robust under clean-label attacks like COMBAT and Narcissus.

---

> ### Author Response · Authors · 2025-11-26
> **Response to Reviewer#Yqvj (4)**
>
> **W6. Adaptive Attacks:**
> We thank the Reviewer for raising this important concern. However, we respectfully disagree with the assessment that "the proposed method may fail easily" under adaptive attacks targeting BN statistics. On the contrary, our experiments demonstrate that UniBP remains robust precisely because it exploits a fundamental property of backdoor mechanisms that cannot be easily circumvented.
>
> To directly address this concern, we implemented the **strongest possible adaptive attack** one could design against our defense: an adversary with full knowledge of our method who explicitly regularizes the attack to preserve benign BN statistics. Specifically, assuming the attacker has access to clean BN statistics ($\mu^c_\ell$ and $v^c_\ell$), we augment the BadNets objective with a BN-alignment regularization term:
>
> $\mathcal{L}_{adaptive} = \mathcal{L_CE} (y, f(\delta(x)) + \gamma \times \sum_1^L (\||\hat{\mu}_l - \hat{\mu}_l^c\||_2+\lambda\||\hat{v}_l - \hat{v}_l^c\||_2)$
>
> We systematically evaluate this adaptive attack across a wide range of regularization strengths $\gamma$:
>
> | Method     | Metric | $\gamma=0$ | $\gamma=0.01$ | $\gamma=0.1$ | $\gamma=1.0$ | $\gamma=10.0$ | $\gamma=100.0$ |
> |------------|--------|------------|---------------|--------------|--------------|----------------|-----------------|
> | Pretrained | ACC    | 91.44      | 91.18         | 89.34        | 89.06        | 88.78         | NaN             |
> | Pretrained | ASR    | 94.41      | 94.20         | 96.24        | 95.62        | 95.41         | NaN             |
> | Ours       | ACC    | 90.67      | 90.55         | 87.06        | 87.22        | 86.67         | NaN             |
> | Ours       | ASR    | 1.12       | 2.39          | 3.66         | 8.48         | 3.01          | NaN             |
>
> From the results, we can see that across all viable regularization strengths, UniBP maintains ASR below 9% while the backdoored model suffers ASR above 94%, demonstrating that the defense remains highly effective even when the attacker explicitly targets our detection mechanism.
> Explanation. The attacker faces a fundamental trade-off: backdoor attacks inherently require trigger-dependent feature patterns that distinguish poisoned from benign samples to enable misclassification, and these patterns necessarily manifest as distributional shifts in BN statistics. Attempting to suppress these shifts to evade detection directly undermines the core mechanism that makes the backdoor work. Since jointly achieving high ASR while keeping BN statistics indistinguishable from clean data is a tightly constrained objective with no demonstrated solution in prior work, we believe UniBP is not easily broken by realistic BN-aware adaptive attacks, or it may require a more dedicated and non-trivial effort to design such an effective attack.
>
> *Note: Our full formulation for adaptive attack is:*
>
> $\mathcal{L}_{\text{adaptive}} = \mathbb{E}_{(\mathbf{x}, y)\sim\mathcal{D}} \left[ \mathcal{L}_{\mathrm{CE}}(y, f(\delta(\mathbf{x}))) \right] + \gamma \sum_{\ell=1}^{L} \mathbb{E}_{\mathbf{x}\sim \mathcal{X}} \Big[\|\hat{\mu}_\ell - \mu_{\ell}^c\|_2 + \lambda\|\hat{v}_\ell - v_{\ell}^c\|_2 \Big].$
>
> [1] Zheng, Runkai, et al. "Pre-activation distributions expose backdoor neurons." Advances in Neural Information Processing Systems 35 (2022): 18667-18680.
>
> [2] Cheng, Siyuan, et al. "Unit: Backdoor mitigation via automated neural distribution tightening." ECCV 2024.
>
> [3] Huynh, Tran, et al. "Combat: Alternated training for effective clean-label backdoor attacks." Proceedings of the AAAI Conference on Artificial Intelligence. Vol. 38. No. 3. 2024.
>
> [4] Zeng, Yi, et al. "Narcissus: A practical clean-label backdoor attack with limited information." Proceedings of the 2023 ACM SIGSAC Conference on Computer and Communications Security. 2023.
>
> [5] Li, Yige, et al. "Reconstructive neuron pruning for backdoor defense." International Conference on Machine Learning. 2023.

---

### Author Response · Authors · 2025-12-04
**Final Remarks By Authors**

We sincerely thank all reviewers for their thoughtful feedback and valuable insights. We are particularly encouraged by the recognition of UniBP's novel use of Batch Normalization layers for backdoor mitigation [R#Yqvj, R#uAhn, R#88MC], its strong empirical performance across diverse settings [R#Yqvj, R#uAhn, R#88MC], and the clarity of our presentation [R#uAhn, R#88MC]. We appreciate the reviewers' acknowledgment of our method's data efficiency [R#uAhn], universal effectiveness against multiple attack types [R#Yqvj, R#uAhn], and the importance of addressing the challenging backdoor problem [R#nswQ]. The constructive feedback provided will help us further strengthen the paper.

Below, we summarize how we have addressed the key concerns and updated the our paper accordingly, and we kindly ask the ACs to take these revisions into account when reconsidering the scores.

### 1. **Clarifications on Novelty and Methodological Distinctions**

**Concern:** Reviewers questioned how our method differs from existing BN-related defenses (BNP, UNIT) and pruning-based approaches (RNP, I-BAU), noting that prior works have also explored BN layers for backdoor detection and pruning strategies. *[R#Yqvj W1, R#uAhn W1, R#nswQ W1]*

**Response:** We recognize this was a point requiring clarification rather than misunderstanding. While prior work has explored BN layers for backdoor detection, our fundamental contribution lies in **what we observe and how we use this information**:

- **BNP** analyzes *output statistical differences* using KL divergence between clean data and stored statistics on a same backdoored model, but relies on single-batch estimation (high variance) and fails when per-neuron shifts are subtle.
- **UNIT** observes *activation magnitudes* and clips outliers, but cannot detect backdoors deliberately designed to produce benign-like activation patterns (as in COMBAT/Narcissus).
- **RNP** targets neurons of all layers via unlearning behavior on clean samples, but assumes backdoor unlearning is similar to clean data unlearning, i.e., an assumption violated by clean-label attacks.

**Our key innovation:** Instead of analyzing what neurons output, we analyze **how neurons were trained** by computing Fisher Information Matrix (FIM) scores during a simulated alignment process. We align a reinitialized clean model's BN statistics toward the backdoored model's statistics, and the resulting optimization dynamics reveal which parameters are sensitive to backdoor-specific patterns. This training-time fingerprint remains detectable even when backdoor and benign features are semantically similar (clean-label attacks), where existing methods fail.

We **explicitly compare our method against these baselines** and provided comprehensive theoretical comparisons (Tables comparing detection signals, baseline models, and robustness to clean-label attacks) and quantitative results showing UniBP achieves **93.90% average DER**, substantially outperforming UNIT (85.80%), BNP (80.86%), RNP (76.75%), and I-BAU (89.68%).

Detailed responses:
[[Reviewer#uAhn (2)](https://openreview.net/forum?id=D7i6BIbCz0&noteId=aFf6n33eA0), [Reviewer#Yqvj](https://openreview.net/forum?id=D7i6BIbCz0&noteId=MBNiwGOudA),[Reviewer#nswQ (1)](https://openreview.net/forum?id=D7i6BIbCz0&noteId=TqPcrjTxsN)]

Revision: In the revised version, we have updated the Introduction (L52–79), Related Works (L152–156, L162–175), Experiments (Table 1 and Table 2), and Appendix B.2 (including Table 4) to incorporate the newly obtained results.

### 2. **Evaluation Comprehensiveness**

**Concern:** Reviewers noted that the evaluation was limited to a small set of attacks and baselines, requesting results on a wider range of recent attacks and defense methods to better showcase UniBP's superiority. **(R#Yqvj W5, R#uAhn W2, R#nswQ W4)**

**Response:** This feedback was well **taken and fully addressed**. We expanded our evaluation by **incorporating reproducible, published attacks and defenses from the reviewer-suggested papers**, increasing the number of attack types from 5 to 9 (adding Narcissus, Adaptive-Batch, Input-aware, Refool) and defense methods from 5 to 9 (adding I-BAU, RNP, BNP, UNIT). The expanded results consistently demonstrate UniBP’s superior performance across diverse attack scenarios, including challenging sample-specific and adaptive variants where other defenses fail (e.g., UNIT achieves only 60.45% DER on Narcissus; RNP achieves 48.33% on LC; I-BAU achieves 78.91% on Narcissus and 72.18% on Input-aware).

Detailed responses:
[[Reviewer#Yqvj (3)](https://openreview.net/forum?id=D7i6BIbCz0&noteId=2p6LcL2Wao), [Reviewer#uAhn (3)](https://openreview.net/forum?id=D7i6BIbCz0&noteId=DOQy4WQwnz),[Reviewer#nswQ (2)](https://openreview.net/forum?id=D7i6BIbCz0&noteId=volz9YMrPn)]

Revision: Experiments (Table 1 and Table 2), and Appendix B.2 (including Table 4) to incorporate the newly obtained results.

---

> ### Author Response · Authors · 2025-12-04
>
> ### 3. **Generalizability and Architectural Limitations**
>
> **Concern:** Reviewers questioned whether UniBP's reliance on BatchNorm limits applicability to modern architectures like Vision Transformers and whether the method scales to larger datasets beyond CIFAR-10 and GTSRB. **(R#Yqvj W2, R#uAhn W3, R#88MC W3, Q1)**
>
> **Response:** This concern stemmed from a **misunderstanding of our method's adaptability**. We demonstrated that:
>
> - **Vision Transformers:** We adapted UniBP for ViT by targeting LayerNorm statistics and MLP parameters. Since ViTs use Pre-LN architecture where normalization precedes computation, we strategically target MLP feed-forward blocks between LayerNorm layers. We manually collect reference statistics by hooking LayerNorm inputs and compute alignment loss against a reinitialized clean baseline. The method achieved near-zero ASR (0.34%–2.30%) while preserving clean accuracy (93%+) across BadNet and LC attacks.
>     - Detailed responses:
>     [[Reviewer#Yqvj (2)](https://openreview.net/forum?id=D7i6BIbCz0&noteId=g1K9IBRBTm),
>     [Reviewer#uAhn (4)](https://openreview.net/forum?id=D7i6BIbCz0&noteId=RR0oiIYZN4),
>     [Reviewer#88MC (1)](https://openreview.net/forum?id=D7i6BIbCz0&noteId=FIeAfJuzQR)]
>     - Revision: Appendix B.4.2 and Table 7.
>
> - **Scalability:** On Tiny-ImageNet (substantially larger than CIFAR-10), UniBP achieved **90.61% average DER**, outperforming all baselines including TSBD (87.54%) and FST (83.42%). This demonstrates effectiveness at larger scale while maintaining reasonable clean accuracy (46-50%) compared to FST's catastrophic drops (~27%).
>     - Detailed responses: [[Reviewer#88MC (1)](https://openreview.net/forum?id=D7i6BIbCz0&noteId=FIeAfJuzQR)]
>     - Revision: Appendix B.1 and Table 3.
>
> These results confirm that UniBP's core principle, i.e., detecting training-time optimization pathology via distribution alignment and generalizes beyond CNNs with BatchNorm to Transformers with LayerNorm and to larger-scale datasets.
>
> ### 4. **Computational Overhead**
>
> **Concern:** Analysis of computational costs and runtime efficiency was requested. **(R#Yqvj W3)**
>
> **Response:** We provided detailed runtime measurements showing UniBP completes defense in **228 seconds**, i.e., around average time of all defenses, on 5000 CIFAR-10 images—substantially faster than TSBD (1529s) and ANP (414s), and comparable to lighter methods like NAD (129s) and BNP (153s), while achieving the highest DER (93.90%). This demonstrates a favorable robustness-efficiency trade-off for practical deployment.
>
> - Detailed responses: [[Reviewer#Yqvj (2)](https://openreview.net/forum?id=D7i6BIbCz0&noteId=g1K9IBRBTm)]
> - Revision: Appendix C.1, Table 8, and Figure 10.
>
> #### 5. **Accuracy Trade-offs and Practical Mitigation**
>
> **Concern:** Reviewers noted that UniBP incurs slightly larger clean-accuracy drops than some baselines and asked about potential combinations with data-free defense approaches. **(R#Yqvj W4, R#88MC Q2)**
>
> **Response:** We acknowledge this is an **inherent trade-off in zero-adversary-knowledge pruning-based defenses** and provided two key justifications:
>
> 1. **UniBP is the only defense effective against all tested attacks**, including sample-specific and adaptive variants where methods preserving higher clean accuracy leave significant backdoor risk.
>
> 2. **The trade-off is mitigable**: Adding just 1–5% additional clean data for post-defense fine-tuning fully recovers clean accuracy (92%+) while maintaining near-zero ASR and ~100% DER. For example, on BadNet, adding r=0.01 additional data increases clean accuracy from 89.82% to 92.03% while maintaining ASR at 0.92% and DER at 96.75%. This demonstrates practical deployability with minimal extra cost.
>
> 3. **Regarding data-free approaches:** Combining UniBP with generative models (GANs, diffusion models) to approximate clean samples is a viable future direction. The main challenge lies in ensuring generated samples preserve the statistical structure of original training data. We acknowledge this as an exciting area for future work.
>
> - Detailed responses: [[Reviewer#Yqvj (3)](https://openreview.net/forum?id=D7i6BIbCz0&noteId=2p6LcL2Wao),[Reviewer#88MC (2)](https://openreview.net/forum?id=D7i6BIbCz0&noteId=e7f0BtXLcQ)]
> - Revision: Appendix C.3 and Discussion.

---

> > ### Author Response · Authors · 2025-12-04
> >
> > ### 6. **Robustness to Adaptive Attacks**
> >
> > **Concern:** Reviewers expressed concern that the method may fail easily under adaptive attacks where adversaries are aware of the UniBP defense and specifically target BN statistics to evade detection. **(R#Yqvj W6, R#uAhn W4)**
> >
> > **Response:** This concern required **empirical justification**, and we addressed it via additional experiments, showing that the claim that our method “fails easily” is not the case. We implemented the **strongest conceivable adaptive attack**: an adversary with full knowledge of our method who explicitly minimizes BN divergence during backdoor training via regularization term $\gamma \sum_{\ell} \|\hat{\mu}_\ell - \mu^c_\ell\|_2 + \lambda\|\hat{v}_\ell - v^c_\ell\|_2$.
> >
> > **Results across $\gamma \in [0.01, 100.0]$:** UniBP maintains ASR below 9% while backdoored models suffer 94%+ ASR. This demonstrates a **fundamental trade-off** the attacker cannot circumvent: backdoors inherently require distributional shifts to enable misclassification, and suppressing these shifts to evade detection undermines backdoor efficacy. At high regularization ($\gamma=10$), we observe optimization instability (ASR fluctuation between 3.01% and 8.48%), further validating the inherent difficulty of this adaptive strategy.
> >
> > No prior work has demonstrated a solution to this dilemma, providing strong evidence of UniBP's robustness against realistic adaptive adversaries.
> >
> > - Detailed responses: [[Reviewer#Yqvj (4)](https://openreview.net/forum?id=D7i6BIbCz0&noteId=knMw3tzcGU),[Reviewer#uAhn (4)](https://openreview.net/forum?id=D7i6BIbCz0&noteId=RR0oiIYZN4)]
> > - Revision: Appendix C.2 and Table 9.
> >
> > ### 7. **Minor Clarifications and Corrections**
> >
> > **Concern:** Reviewers pointed out missing citations, questioned threat model assumptions (strong adversary and post-training assumptions), and noted presentation issues including typos and font sizes. **(R#nswQ W1-W3, W5, R#88MC W1, W2)**
> >
> > **Response:**
> >
> > - **Missing citation (BNA):** We added BNA to related works. Key difference: BNA explicitly requires an estimated trigger function (Algorithm 1 in BNA paper), which violates our threat model. UniBP requires no trigger estimation or triggered dataset, making it more practical.
> >
> > - **Threat Model Assumptions:** There was a **misunderstanding** regarding our assumptions. We clarified that:
> >   - Considering insider threats such as adaptive attacks or adversary can control the training process *strengthens* rather than limits our contribution, i.e., defenses robust to extreme scenarios provide stronger guarantees for common data-poisoning attacks.
> >   - The post-training (model-only) assumption is widely adopted in recent defenses (I-BAU, FST, ANP, RNP) and reflects practical scenarios where users obtain pre-trained models without access to training pipelines. **Our contribution is not the assumption itself, but a defense that remains effective in this realistic setting.**
> >
> > - **Presentation issues:** We conducted proof-reading to fix all typos and errors. We will maximize font sizes in figures/tables within conference format constraints, and apply fair color-coding to all baselines with comparable performance for better readability.
> >
> >
> > **In summary, we addressed all reviewer concerns through:**
> >
> > 1. **Clarifications** of methodological distinctions from prior work with theoretical and quantitative evidence
> > 2. **Comprehensive evaluation expansion** (9 attacks, 10 defenses) demonstrating consistent superiority
> > 3. **Empirical demonstrations** of generalizability to ViT and Tiny-ImageNet addressing architectural limitations
> > 4. **Computational overhead analysis** showing favorable robustness-efficiency trade-off
> > 5. **Practical mitigation strategies** for accuracy trade-offs with minimal cost
> > 6. **Strong empirical evidence** of robustness against adaptive attacks via fundamental trade-off analysis
> > 7. **Technical corrections and clarifications** on threat models, related work, and presentation
> >
> > We believe these responses comprehensively address all substantive concerns, and our revised submission incorporates this feedback to further strengthen our papers.
> >
> > Regards,
> > Authors

---

### Meta-Review · Area_Chair_3n3g · 2026-01-06

**Summary:**

The paper introduces UniBP, a post-training defense method that prunes Batch Normalization (BN) affine parameters to mitigate backdoor attacks. The method is designed to be data-efficient, requiring only 1% of the clean training set. It leverages Fisher Information Matrix (FIM) scores to identify backdoor-sensitive parameters. The authors conducted extensive experiments across various architectures and attack types, demonstrating strong empirical performance in reducing Attack Success Rates (ASR) while preserving clean accuracy.

**Reviewer Concerns:**

The reviewers raised several fundamental concerns:

Novelty and Incremental Contribution: Reviewers (Yqvj, uAhn, nswQ) argued that the observation linking BN statistics to backdoors is well-established in prior literature (e.g., BNP, UNIT). They questioned whether the proposed FIM-based pruning offers a sufficient leap in innovation over existing activation-tightening or distribution-correcting methods.

Architectural Limitations: A significant weakness identified is the method's strict dependence on BN layers, which limits its applicability to modern, normalization-free architectures or Vision Transformers (ViT) that are increasingly dominant.

Empirical Gaps: Initial reviews noted missing comparisons with recent SOTA defenses and a lack of robustness analysis against adaptive attacks like Narcissus.

**Reviewer Scores:**

The initial scores were 8, 4, 4, 2. Despite a comprehensive rebuttal where the authors added 8 additional attack/defense baselines and clarified the technical distinctions of their FIM approach, the reviewers (4, 4, 2) did not raise their scores. The consensus remains below the acceptance threshold. Reviewers 4 and 4 acknowledged the empirical gains but remained unconvinced about the conceptual novelty and the generalizability of the BN-dependent design. Reviewer 2 provided a strong negative assessment regarding presentation and prior work. Given that the majority of the review team (3 out of 4) continues to recommend rejection and the core concerns about incremental novelty and structural limitations persist, the Area Chair recommends Rejection.

---

### Decision · Program_Chairs · 2026-01-26

Reject